# Transition Machine Teaching

## Abstract

Machine teaching endeavors to minimize the divergence between the teacher and learner within the model parameter space to facilitate the identification of critical data. However, conventional methods for achieving this typically rely on closed-form function operations, which often introduce inconsistencies in parameter spaces. Theoretically, these inconsistencies diminish the interpretability of the learner and reduce it to a black-box system. This paper advocates a paradigm shift in machine teaching, transitioning from *conventional direct parameter space matching* toward a more nuanced approach focused on *aligning teacher's parameter space with learner's data distribution.* Specifically, we propose a novel framework for projecting the learner's data distribution onto the gradient space of the converged model. This projection facilitates the quantification of uncertainty within the gradient transition space, enabling the identification and elimination of redundant distributions while sampling the essential coverage of the trust distribution. Utilizing the inherent unbiased properties of the teacher's parameter space, we further propose regulatory constraints to systematically guide the optimization of the learner's data distribution. Theoretical analysis and comprehensive results conducted across diverse scenarios substantiate the efficacy of this transition.

## 1 Introduction

Machine teaching focuses on optimizing the selection of training examples to enhance model performance. It is often framed as the inverse problem of machine learning (Zhu et al., 2018), where the goal is to determine the most effective training set for a specific learning algorithm and target model. This framework has been applied in diverse domains, including inverse reinforcement learning (Ho et al., 2016; Brown & Niekum, 2019), curriculum learning (Zhou & Bilmes, 2018), education-driven systems (Piech et al., 2015; McNichols et al., 2023; Phung et al., 2023), data annotation and augmentation for large-scale models (He et al., 2023), and self-supervised reasoning (Zelikman et al., 2022; Madaan et al., 2024).

Substantial research in machine teaching usually investigates the iterative interaction between teachers and learners (Liu et al., 2018; Dasgupta et al., 2019; Zhang et al., 2023; Fallat et al., 2023; Sucholutsky et al., 2024), with a primary focus on optimizing teaching examples through an in-depth understanding of the learner's cognitive state. These theoretical contributions, alongside their practical applications, have played a crucial role in advancing the field.

**Theoretical teaching challenges.** In machine teaching, learners typically rely on a closed-form operation on model parameters (Liu et al., 2017; Fallat et al., 2023), such as $\|\theta_T - \theta_S\|_2^2 \leq \epsilon$, to quantify the effectiveness of selected data, thereby minimizing the divergence between the teacher's and learner's parameters. This methodology is effective under ideal conditions where the teacher and learner models exhibit identical structures, perfect alignment, and congruent data distributions. However, in practical applications, it is far more common to encounter situations in which model structures are heterogeneous, models, while structurally similar, are difficult to align, or there is a substantial discrepancy in the data distributions between the teacher and learner. These challenges render the alignment-based measure, $\|\theta_T - \theta_S\|_2^2$, impractical in high-dimensional parameter spaces.

**Gradient transition.** Given these challenges, an alternative approach investigates how individual samples interact with the model through gradients. As gradients reflect the sensitivity of the loss function to model

parameters, they offer a natural way to assess the influence and informativeness of each sample. They are also widely used in interpretability (Koh & Liang, 2017b; Ghorbani et al., 2019; Pruthi et al., 2020; Szolnoky et al., 2022) and uncertainty quantification (Wang et al., 2022; Daheim et al., 2023; Chhabra et al., 2024). In the typical machine teaching scenarios, where the teacher model and the learner's dataset are known, transiting the gradients of teacher model to learner's data provides an effective inference metric that eliminates the need for feedback from parameter matching paradigms.

**Transition machine teaching.** In this paper, we propose a new method in which the learner's samples require only a single forward and backward pass through the teacher model. Specifically, we transit the learner's samples to the gradient space of converged checkpoints in the teacher's training process to obtain an integrated distribution of data gradient information quantification scores. This step ensures responsiveness to out-of-pattern data. Once the gradient quantification scores are deemed sufficiently effective, we then prune redundant distributions based on the distrusted score area and finally extract a representative transition set based on the principles of coreset coverage (Zheng et al., 2022). Notably, our method does not require multiple feedback interactions with the learner, effectively avoiding cumulative errors that may arise from mismatches in parameter and data distributions. In summary, our contributions are as follows:

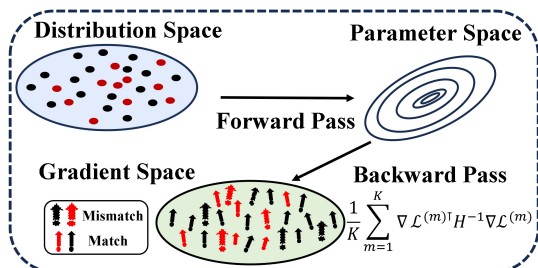

Figure 1: Transition machine teaching conducts a forward operation on the learner's data within the parameter space of the teacher model. Subsequently, it uses teaching influence function to map these influence scores into the transition gradient space.

- **Structured cross-space alignment.** We propose a method that sidesteps conventional parameter space matching, removing the reliance on parameter space feedback for sample selection. Instead, it emphasizes implicit cross-space alignment between parameter and data distribution spaces, mitigating teaching bias accumulation from distribution mismatches and structural discrepancies.

- **Theoretical analysis of dynamic bias.** We analyze the dynamic bias arising from the mismatch between the latent data distribution of the teacher model and that of the learner. Using linear mode connectivity (Frankle et al., 2020; Entezari et al., 2022), we analyze how biases in closed-form operations (e.g., linear interpolation) within the parameter space are influenced by inconsistent parameter permutations and exacerbated by teaching biases due to gradient shifts from data distribution mismatches. Furthermore, we apply distributionally robust theory to quantify the generalization error bounds resulting from these teaching biases.

- **Transition paradigm and validation.** Using the teaching influence function, we are able to establish a connection between parameters and data points, thereby creating a cross-operation between parameters and data distributions. This operation utilizes the teacher's parameter space to impose effective normative constraints on the learner's dataset, demonstrating enhanced robustness and adaptability. We conduct experimental validation across multiple datasets and various data distribution scenarios, including imbalanced and noisy data situations.

This paper is structured as follows: Section 2 reviews relevant literature; Section 3 elaborates on the fundamental concepts of machine teaching to lay a theoretical foundation; Section 4 explores the motivations for the proposed transition and conducts comprehensive theoretical analysis to justify it; Section 5 details transition machine teaching, including its core principles, implementation framework and key operational steps; Section 6 presents experimental analysis to validate the method's effectiveness; Finally, Section 7 summarizes the work's main contributions, reflects on limitations and outlines future research directions.

## 2 Related Work

**Machine teaching.** In recent years, research on machine teaching has largely focused on the interactions between teachers and learners. To accelerate learner convergence, (Liu et al., 2017) proposed Iterative Machine

Teaching, where teachers dynamically distribute examples based on iterative feedback from learners. In a scenario where teaching features differ and learners are treated as black boxes, (Liu et al., 2018) introduced a method for teachers to actively query the learner's status. (Dasgupta et al., 2019) emphasized the necessity of interacting with black box learners, enabling teachers to effectively find the optimal set of teaching examples. (Cicalese et al., 2020) refined some existing results from (Dasgupta et al., 2019), studying various new variants of learners under different performances. To address the high time complexity of (Liu et al., 2017), (Xu et al., 2021) proposed Locality Sensitive Teaching. (Liu et al., 2021) further introduced an input example label synthesis teaching framework to avoid the high costs of example selection. In research by (Wang et al., 2021), teachers aligned the steepest descent direction of the teaching set risk with that of the empirical risk of the entire dataset.

In other aspects, (Zhou & Bilmes, 2018) emphasized the adaptive selection of subsets based on difficulty and diversity during training. (Brown & Niekum, 2019) proposed finding the minimum number of examples needed to identify reward equivalence classes in inverse reinforcement learning. (Sanchez et al., 2022) explored the uncertainty in machine teaching within human-machine interaction, finding that participants using more diverse training sets better understood uncertainty. (Wu et al., 2024) studied machine teaching in discrete domains, modeling it as a combinatorial optimization problem. (Sucholutsky et al., 2024) find that representational alignment enhances learning outcomes and designing a classroom matching program to optimize teacher-student pairing.

**Influence function.** The research surge on influence functions originated from (Koh & Liang, 2017a), which attributes the value of training data by assessing the impact of adding or removing data points on model prediction performance. Due to the complexity of computing the Hessian matrix and its inverse, (Pruthi et al., 2020; Bae et al., 2022; Grosse et al., 2023; Schioppa et al., 2022) approach the problem from different angles to avoid direct Hessian computation. (Bae et al., 2022) introduced the Gauss-Newton Hessian, while (Grosse et al., 2023) proposed Eigenvalue-corrected Kronecker-Factored Approximate Curvature (EK-FAC) for hierarchical block approximation. (Schioppa et al., 2022) extracted the effective dimensions of the Hessian matrix. TracIn (Pruthi et al., 2020) simplifies influence by setting the Hessian matrix as the identity matrix, tracking the cumulative change in loss through the dot product of gradients at multiple checkpoints. Additionally, (Kong et al., 2021) uses influence functions to locate harmful samples and relabel them, while (Jung et al., 2024) employs self-influence to identify biased samples and correct biased models.

## 3 Conceptual Preliminaries

**General form of machine teaching.** Let $\boldsymbol{\Theta}$ be the parameter space, and $\theta_T^* \in \boldsymbol{\Theta}$ be the target specified by the teacher. The learner is modeled as an algorithm $\mathcal{A} : \mathbf{D} \to \boldsymbol{\Theta}$, where $\mathbf{D}$ denotes the data distribution space, typically represented as a dataset $\mathcal{D} = \{(x_i, y_i)\}_{i=1}^n$. The goal of machine teaching (Zhu et al., 2018) is to find the minimal dataset $\hat{\mathcal{D}}_S \subseteq \mathcal{D}$ such that: $\hat{\mathcal{D}}_S = \arg\min_{\mathcal{D}_S \in \mathcal{D}} |\mathcal{D}_S|$ s.t. $\mathcal{A}(\hat{\mathcal{D}}_S) = \theta_T^*$. $|\mathcal{D}_S|$ denotes the size of the training set. In practice, a performance criterion is often used, requiring the learned model $\hat{\theta}_S = \mathcal{A}(\hat{\mathcal{D}}_S)$ to be within a threshold $\epsilon$ of the target $\theta_T^*$: $\|\hat{\theta}_S - \theta_T^*\|_2^2 \le \epsilon$. The optimization objective can then be defined as:

$$\min_{\hat{\mathcal{D}}_S, \hat{\theta}_S} \left\| \hat{\theta}_S - \theta_T^* \right\|_2^2 + \eta |\hat{\mathcal{D}}_S|, \text{ s.t. } \hat{\theta}_S = \underset{\theta_S \in \Theta}{\arg\min} \sum_{(x,y) \in \hat{\mathcal{D}}_S} \mathcal{L}(f\langle \theta_S, x \rangle, y) + \lambda \|\theta_S\|_2^2, \tag{1}$$

where $\mathcal{L}(,)$ is loss function, $f$ denotes the learning model, and $\lambda$ is a regularization parameter.

**Iterative machine teaching** (IMT) (Liu et al., 2017; 2018) offers a practical solution to machine teaching by progressively optimizing the dataset through multiple interactions. Let $\mathcal{D}_S^{t*} = \{d_1, \dots, d_t\}$ be the dataset accumulated up to step $t$, with the learner's state $\hat{\theta}_S^t = \mathcal{A}(\mathcal{D}_S^{t*})$. At step $t+1$, the teacher selects $d_{t+1}$ to minimize the error between the updated learner and the target $\theta_T^*$:

$$d_{t+1} = \arg\min_{d \in \mathcal{D}_S} \left\| \hat{\theta}_S^{t+1} - \theta_T^* \right\|_2^2 = \arg\min_{(x,y) \in \mathcal{D}_S} \left\| \hat{\theta}_S^t - \eta_t \frac{\partial \mathcal{L}(f\langle \theta, x \rangle, y)}{\partial \theta} - \theta_T^* \right\|_2^2. \tag{2}$$

**Nonparametric iterative machine teaching** (NIMT) (Zhang et al., 2023) extends the iterative machine teaching method by optimizing the dataset through prediction divergence maximization. In each round $t$ of the total epochs $T$, the data loader is divided into batches $\mathbb{B}_1, \ldots, \mathbb{B}_K$ based on the batch size. Let $\mathcal{D}_S^{m*} = \{d_1, \ldots, d_m\}$ be the accumulated dataset up to the $m$-th batch. In the $(m+1)$-th block $\mathbb{B}_{m+1}$, the teacher selects the next samples $d_{m+1}$ based on prediction divergence maximization:

$$d_{m+1} = \arg \max_{d \in \mathbb{B}_{m+1}} \left| f_{\hat{\theta}_{S_m}^t}(\mathbb{B}_{m+1}) - f_{\theta_T^*}(\mathbb{B}_{m+1}) \right|. \tag{3}$$

This teaching method shares a similar foundation with the method proposed in this paper. It selects teaching sets based on prediction divergence rather than relying on parameter disagreement control.

**Teaching influence score.** Influence function explores how the optimal model parameters $\theta^*$ change when a training sample $z = (x, y)$ is upweighted by a weight of $\epsilon$: $\theta_{\mathbf{z}, \epsilon}^* = \arg \min_{\theta \in \mathbb{R}^D} \frac{1}{N} \sum_{i=1}^{N} \mathcal{L}([f(x_i), y_i]; \theta) + \epsilon \mathcal{L}(f(x, y); \theta)$. Influence of the sample $z$ on the parameters $\theta^*$ can be computed using the Implicit Function Theorem as shown in (Koh & Liang, 2017b; Bae et al., 2022), through a second-order Taylor expansion: $\mathcal{I}_{\theta^*}(z) = \frac{d\theta^*}{d\epsilon}\Big|_{\epsilon=0} = -H_{\theta^*}^{-1} \nabla_\theta \mathcal{L}(z; \theta^*)$, where $H_{\theta^*} = \nabla_\theta^2 \mathcal{L}(f; \theta^*)$ is the Hessian matrix of the loss with respect to the $\theta^*$. Since $\mathcal{I}_{\theta^*}(z)$ is generally difficult to interpret directly, researchers (Grosse et al., 2023) typically compute its influence on measurable quantities $\nabla_\theta \mathcal{F}(\theta^*)$. For a training sample $z$, the teaching influence score is defined as the gradient of the teaching objective $\mathcal{F}$ with respect to the model parameters: $\nabla_\theta \mathcal{F}(\theta^*) = \nabla_\theta \mathcal{L}(z; \theta^*)$. By the chain rule, we obtain

$$\mathcal{I}_{\mathcal{F}}(z) = \frac{d\mathcal{F}(\theta^*)}{d\epsilon} = \frac{\partial \mathcal{F}(\theta^*)}{\partial \theta^*} \frac{\partial \theta^*}{\partial \epsilon} = \nabla_\theta \mathcal{F}(\theta^*)^\top \mathcal{I}_{\theta^*}(z) = \nabla_\theta \mathcal{L}(z; \theta^*)^\top \mathcal{I}_{\theta^*}(z). \tag{4}$$

Thus, the teaching influence function explicitly links the training sample and the learned parameters. In our method, this is demonstrated in Figure 1, and a detailed discussion is provided in Section 5.

## 4 Motivation and Analysis

**Motivation. Motivation.** During the teaching process, dynamic bias arises from approximate convex optimization. Building on the iterative nature of teaching processes, Section 4.1 examines the adverse impact of data distribution mismatch between teacher and learner models on their learning dynamics. Further analysis of Proposition 4.1 reveals that directly optimizing their disagreements—whether in the parameter or function space—will exacerbate dynamic bias. This key insight lays the foundation for our proposed Transition Machine Teaching method: unlike conventional approaches that may introduce implicit, hard-to-quantify biases via parameter or function optimization, our method avoids such optimization entirely during the teaching process.

**Theoretical analysis.** To theoretically validate the feasibility of this method and address the limitations of existing strategies, we present the following analysis. In Section 4.2, we clarify that standard domain generalization and distribution shift methods are not applicable due to the unknown training distribution of the teacher. Instead, we use distributionally robust optimization: we treat the learner's distribution as a perturbation of the unseen teacher distribution and define an uncertainty set via divergence measures (e.g., Hellinger distance). We then derive worst-case generalization bounds under these shifts through Hellinger analysis, providing a practical theoretical basis for Transition Machine Teaching's cross-space operations and their robustness to dynamic bias.

**Transition scheme.** Building on the above theoretical foundation, we further propose a concrete implementation framework to isolate and mitigate dynamic bias. We introduce a novel framework that operates across the teacher's parameter space and the learner's distribution, explicitly isolating and smoothing these biases. In short, typical teaching requires homomorphic operations within an aligned teaching dynamic in parameter or function space, while heteromorphic operations are needed across spaces to address distribution misalignment during the teaching dynamics.

### 4.1 Teaching bias in optimization

From approximate convex optimization, gradient deviations caused by distribution mismatch inevitably introduce dynamic teaching bias. In real-world scenarios, the learner's data distribution often diverges from the teacher's intended distribution. This divergence induces gradient deviation and causes misalignment between the learner and the teacher target in parameter or function space. Moreover, in high-dimensional parameter spaces, the non-convex nature of the optimization process further amplifies these alignment errors (Bernstein et al., 2020; Yang et al., 2023). The combined effect of gradient deviation and high-dimensional metrics ultimately leads to suboptimal sample selection. Leveraging Linear Mode Connectivity (LMC) (Frankle et al., 2020; Entezari et al., 2022), we provide a detailed analysis of misalignment errors and sample selection bias in iterative teaching process below.

**Insight 4.1.** *LMC Insights into Teaching Optimization.* If $\theta_1$ and $\theta_2$ are convergent models with different initial values and different training data orders. LMC [1] loss barriers $\mathcal{L}(f_{\lambda\theta_1+(1-\lambda)\theta_2}) - [\lambda\mathcal{L}(f_{\theta_1}) + (1-\lambda)\mathcal{L}(f_{\theta_2})]$ provides a new perspective for model merging and analyzing linear operations involving $\|\theta_T - \theta_S\|_2^2$. Such loss barriers can be effectively mitigated through strategies including activation alignment, weight matching, permutation application (Ainsworth et al., 2023), or specific layer integration (Adilova et al., 2024).

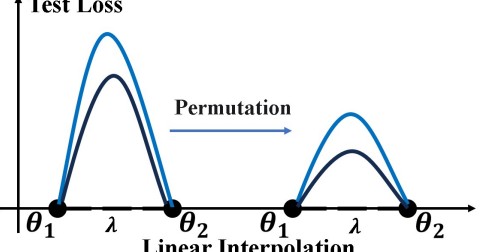

Figure 2: Linear interpolation between two structurally identical models (Frankle et al., 2020) results in a significant loss barrier, while (Ainsworth et al., 2023) effectively reduces test loss through model permutation alignment.

Building on these insights from the LMC framework, the subsequent proposition demonstrates how linear operations in the parameter or function space further amplify the sample selection bias caused by distribution mismatch.

**Proposition 4.1.** *In iterative teaching process, the learner's candidate dataset is $\mathcal{D}_S \sim \mathcal{P}$, and the teacher's goal is to enable the learner to learn a model that is functionally close to the $\theta_T^*$. For the learner, teacher dataset $\mathcal{D}_T \sim \mathcal{T}$ is typically a black box. The selection process for each batch of samples $(x, y)$ follows the procedure in Eq. 2 (or Eq. 3).* **Thus, we can conclude that in nonlinear composite structures such as neural networks, when there is a distribution mismatch ($\mathcal{P} \neq \mathcal{T}$), the gradient $\nabla_\theta \mathcal{L}(f_{\hat{\theta}_S^t}(x), y)$ on $\mathcal{P}$ can lead to an update direction for the learner model that deviates from $\theta_T^*$. Based on LMC theory, the limitations of the high-dimensional metric $\|\cdot\|_2^2$ are further exacerbated in cases of distribution mismatch, amplifying the sample selection bias.**

The proof of Proposition 4.1 is in Appendix A.1. In summary, the combination of approximate convexity optimization and distribution mismatch leads to high-dimensional metrics bias and gradient bias that fail to measure parametric or functional equivalence, further exacerbating sample selection errors. This motivates us to shift from homomorphic operations within a single space to heteromorphic operations across multiple spaces.

### 4.2 Teaching robustness in distribution

Given the teaching bias in optimization dynamics, the robustness of the teaching scheme is appealing. This consideration motivates an investigation into the generalization performance under distribution mismatch between the teacher and the learner. To this end, we employ distributionally robust optimization theory to analyze the generalization error induced by such misalignments. However, a core assumption in machine teaching is that the training data distribution of the teacher is typically unknown. In contrast, many domain generalization and distribution shift theories require both source domain samples $\widehat{\mathcal{S}}$ and target domain samples $\widehat{\mathcal{T}}$ to compute distributional divergence and VC dimension (Ben-David et al., 2010; Zhang et al., 2019), which reduces their applicability within the machine teaching related framework.

---

[1]LMC possesses inherent properties associated with convex combination optimization.

Distributionally robust theory effectively analyzes changes in data distribution under perturbations. Specifically, we use the approach from (Weber et al., 2022), combining probability metrics like Hellinger distance with statistical characteristics of the loss function (e.g., expectation, variance) to quantify the impact of distributional perturbations on model performance. Compared to metrics like KL divergence, the Hellinger distance $H_D$ remains finite even when the support sets of the distributions don't fully overlap, thus avoiding issues of numerical instability such as divergence to infinity.

**Remark 4.1.** Theoretically, machine teaching assumes the learner's original data distribution $\mathcal{P}$ is roughly the same as the teacher's original distribution $\mathcal{T}$, i.e., $\mathcal{P} \approx \mathcal{T}$. But in practice, the learner often follows a different, perturbed distribution $\mathcal{Q}$, which usually does not match $\mathcal{T}$. This mismatch causes the gradient updates to drift away from the parameter space tied to $\mathcal{T}$.

In the above statement, we note that classical generalization theory no longer applies to the unique data distribution assumptions in machine teaching. Therefore, we introduce a distributed robust generalization theory that accounts for data distribution perturbations, which measures distribution shifts through the Hellinger distance. Additionally, in Remark 4.1, we formalize the typical data distribution assumptions in machine teaching. Based on this theoretical groundwork, we utilize Hellinger analysis to specifically examine the generalization error introduced by the accumulated teaching bias during the machine teaching process.

**Hellinger analysis.** Let $H_D(\mathcal{Q}, \mathcal{P})$ characterize the learner's heterogeneous about distribution shift, based on Lemma A.1 in Appendix A.2, it transforms this distribution heterogeneous by examining its learning loss $\mathcal{L}$ over $\mathcal{P}$. This loss is approximately transformed into the expectation $\mathbb{E}_{\mathcal{P}}$ and variance $\mathbb{V}_{\mathcal{P}}$ through a polynomial composite relation, represented as $\mathcal{B}_{\mathcal{L}}(\epsilon, \mathcal{P})$. Ultimately, this leads to the teacher's approximation in a polynomial composite relation, given that $\mathcal{P}$ is close to $\mathcal{T}$ of Remark 4.1. Thus, the learner's self-heterogeneity can be effectively bounded by the teacher distribution as:

$$\forall \mathcal{Q} : H_D(\mathcal{Q}, \mathcal{P}) \leq \epsilon \rightarrow \mathbb{E}_{(x,y)\sim\mathcal{Q}}[\mathcal{L}(h_\theta(x), y)] \leq \mathcal{B}_{\mathcal{L}}(\epsilon, \mathcal{P}) \approx \mathcal{B}_{\mathcal{L}}(\epsilon, \mathcal{T}), \tag{5}$$

where $h_\theta$ represents the learner's model, and $\mathcal{B}_{\mathcal{L}}(\epsilon, \mathcal{P}) = \mathbb{E}_{\mathcal{P}} + 2\lambda_\epsilon[\mathbb{V}_{\mathcal{P}}] + \Delta_\epsilon^u(\epsilon, M, \mathbb{E}_{\mathcal{P}}, \mathbb{V}_{\mathcal{P}})$ denotes the distribution measure. Detailed proof of Inequal. 5 can be found in Appendix A.2.

In machine teaching, by using our transition machine teaching method, the selected efficient data subset for the learner will make $\mathcal{D}_S'$ move away from the biased data distribution $\mathcal{Q}$ and approach the learner's ideal distribution $\mathcal{P}$, $\epsilon$ decreases, and the generalization error bound tightens accordingly. At this point, replacing $\mathcal{Q}$ with $\mathcal{D}_S'$, we have $H_D(\mathcal{D}_S', \mathcal{P}) = \epsilon_1 \leq \epsilon$. Through Inequal. 5, the generalization error bound will also be further tightened with $\epsilon_1$:

$$\mathbb{E}_{\mathcal{D}_S'}[\mathcal{L}(\theta)] \leq \mathbb{E}_{\mathcal{P}}[\mathcal{L}(\theta)] + \lambda_{\epsilon_1}\mathbb{V}_{\mathcal{P}}^{1/2}[\mathcal{L}(\theta)] + \Delta_{\epsilon_1}^u \leq \mathbb{E}_{\mathcal{P}}[\mathcal{L}(\theta)] + \lambda_\epsilon\mathbb{V}_{\mathcal{P}}^{1/2}[\mathcal{L}(\theta)] + \Delta_\epsilon^u. \tag{6}$$

## 5 Transition Machine Teaching

In Section 4.1, we uncovered the mechanism by which dynamic teaching bias arises. In Section 4.2, using distributional robustness analysis, we further demonstrated that this dynamic bias amplifies the upper bound on generalization error. Consequently, as iterations proceed, the distribution of the learner's parameters becomes increasingly unreliable due to accumulated bias.

To address this, we employ cross-space teaching between the teacher's parameter space and the learner's distribution space. This heterogeneous operation, guided by an unbiased teaching model, is structurally designed to isolate bias. To make this bias explicit, we leverage the teaching influence function (Eq. 4), which links model parameters to data distributions. We thus define the cross-space transition computation method for **Transition**

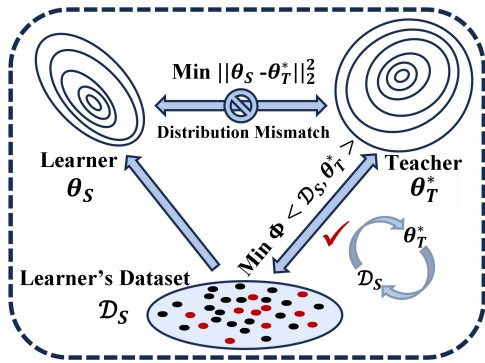

Figure 3: Optimizing teaching with same space becomes challenging in the presence of a distribution mismatch.

**Machine Teaching** as follows:

$$\min_{\hat{\mathcal{D}}_S, \hat{\theta}_S} \sum_{z \in \hat{\mathcal{D}}_S} \Phi(z; \theta_T^*) + \eta \cdot |\hat{\mathcal{D}}_S|, \text{ s.t. } \hat{\theta}_S = \underset{\theta_S \in \Theta}{\operatorname{argmin}} \sum_{(x,y) \in \hat{\mathcal{D}}_S} \mathcal{L}(f\langle \theta_S, x \rangle, y), \ \hat{\mathcal{D}}_S \subseteq \mathcal{D}_S, \quad (7)$$

where $\Phi(z; \theta_T^*) = \nabla_\theta \mathcal{L}(z; \theta_T^*)^\top H_{\theta_T^*}^{-1} \nabla_\theta \mathcal{L}(z; \theta_T^*)$, with $z \in \mathcal{D}_S$. $\Phi(z; \theta_T^*)$ represents the implicit function score of the sample $z$ in the gradient space. The optimization of $\hat{\theta}_S$ is consistent with Eq. 1. Next, we will provide a detailed overview of the algorithmic process of transition machine teaching.

**Teacher's forward and backward propagation.** The converged checkpoint of the teacher model is denoted as $\theta_T^*$. Given the learner's dataset $\mathcal{D}_S = \{(X_S^{(i)}, Y_S^{(i)})\}_{i=1}^N$, we run the teacher's checkpoint on $\mathcal{D}_S$ with one forward pass and one backward pass, without updating the learner's parameters $\theta_S$. For the teacher model checkpoint $\theta_T^*$, a forward pass is performed as $\hat{Y}_T^{(i)} = f_T(X_S^{(i)}, \theta_T^*)$, where $f_T$ represents the forward teacher model. The loss for each data point $i$ is $\mathcal{L}_i = \mathcal{L}(\hat{Y}_T^{(i)}, Y_S^{(i)})$. The gradient for backpropagation is calculated as $\nabla_{\theta_T^*} \mathcal{L}_i = \nabla_{\hat{Y}_T^{(i)}} \mathcal{L} \cdot \nabla_{\theta_T^*} \hat{Y}_T^{(i)}$. For a mini-batch $\mathbb{B}$, the batch gradient is given by $\nabla_{\theta_T^*} L_{\mathbb{B}} = \frac{1}{|\mathbb{B}|} \sum_{j \in \mathbb{B}} \nabla_{\theta_T^*} \mathcal{L}(\hat{Y}_T^{(j)}, Y_S^{(j)})$, where $|\mathbb{B}|$ denotes the batch size.

**Quantifying gradient transition distribution.** In Eq. 7, we establish an explicit connection between model parameters and training samples through cross-operation. Under the assumptions of MT, Eq. 7 is inapplicable because computing the inverse Hessian requires access to the teacher's training data, which is assumed to be unavailable. The Hessian-free method, TracIn (Pruthi et al., 2020), provides us with insights. We thus set the Hessian matrix as the identity matrix $I$, and define the transition gradient quantification of data point $i$ at the converged checkpoint as follows:

$$\mathcal{I}_{\text{Quant}} = \nabla_{\theta_T^*} \mathcal{L}(\hat{Y}_T^{(i)}, Y_S^{(i)})^\top \nabla_{\theta_T^*} \mathcal{L}(\hat{Y}_T^{(i)}, Y_S^{(i)}). \quad (8)$$

It shares a consistent kernel with the self-influence function, thus being labeled as the **teaching influence score** in this paper, which reduces computational complexity compared to Eq. 7. To eliminate the randomness caused by a single set of weights and obtain robust results, we typically use several converged teacher checkpoints: $\mathcal{I}_{\text{Quant}} = \frac{1}{K} \sum_{m=1}^K \nabla_{\theta_T^{(m)*}} \mathcal{L}(\hat{Y}_T^{(m)}, Y_S^{(i)})^\top \nabla_{\theta_T^{(m)*}} \mathcal{L}(\hat{Y}_T^{(m)}, Y_S^{(i)})$, where $K$ denotes the number of checkpoints.

**Transition teaching set with coverage selection.** Using Eq. 8, we first compute the transition gradient quantification $\mathcal{I}_{\text{Quant}}$ for the learner's dataset $\mathcal{D}_S$. Then, through a mismatch pruning ratio $\beta$: $\mathcal{I}_{\text{Quant}} \to \mathcal{I}'_{\text{Quant}} \to \mathcal{D}'_S$, and given a subset rate $\alpha$, we select a subset $\mathcal{D}_{\text{opt}} \subseteq \mathcal{D}'_S$ to maximize the accuracy of the model trained on this subset. $\mathcal{I}'_{\text{Quant}}$ is the transition gradient quantification score obtained by pruning the mismatches from $\mathcal{I}_{\text{Quant}}$, and mapped to the dataset $\mathcal{D}'_S$ via indices. $\mathcal{D}_{\text{opt}}$ denotes the optimal subset filtered hierarchically from $\mathcal{D}'_S$. Thus, this selection process is formulated as the following optimization problem:

$$\min_{\mathcal{D}_{\text{opt}} \subseteq \mathcal{D}'_S : \frac{|\mathcal{D}_{\text{opt}}|}{|\mathcal{D}'_S|} \leq 1 - \alpha} \mathbb{E}_{(x,y) \sim \mathcal{D}_{\text{opt}}}[\mathcal{L}(x, y; \theta_{\mathcal{D}_{\text{opt}}})], \text{ s.t. } \mathcal{D}'_S \leftarrow \text{Prune}_\beta\left(\mathcal{D}_S; \mathcal{I}_{\text{Quant}}(\cdot; \theta_T^*)\right), \quad (9)$$

where $\mathcal{L}$ is the loss function and $\theta_{\mathcal{D}_{\text{opt}}}$ are the model parameters trained on $\mathcal{D}_{\text{opt}}$. We select $\mathcal{D}'_S$ by retaining the $(1 - \beta)|\mathcal{D}_S|$ data points in $\mathcal{D}_S$ with the smallest transition gradient quantification $\mathcal{I}_{\text{Quant}}(x, y; \theta_T^*)$. The entire process can be summarized by the sequence $\mathcal{D}_S \to \mathcal{I}_{\text{Quant}} \to \mathcal{I}'_{\text{Quant}} \to \mathcal{D}'_S \to \mathcal{D}_{\text{opt}}$: using transition gradient quantification as a key intermediate step, the method prunes data lying outside the gradient transition distribution and incorporates the remaining samples into the optimal transition teaching set. The generalization error bound will also be adjusted as in Eq. 6.

In typical practice of machine teaching, we find that selecting a subset of the learner's data often leads to a sharp decline in experimental performance. To ensure good performance of the transition teaching set selection process at a high pruning rate $\alpha$, we employ a density-based distribution cover sampling method

(Zheng et al., 2022). This method draws on the principles of stratified sampling. By allocating the sampling budget based on different information score levels, it ensures better coverage even at high pruning rates. Finally, we summarize the aforementioned transition machine teaching process as Algorithm 1.

**Algorithm 1: Gradient-Coverage Teaching-Set Selection.** Given a labelled pool $\mathcal{D}_S$ and a converged teacher, we first quantify the teaching value of each instance by its squared gradient norm $\mathcal{I}_{\text{Quant}}^{(i)} = \nabla_{\theta_T^*}\mathcal{L}(\hat{Y}_T^{(i)}, Y_S^{(i)})^\top \nabla_{\theta_T^*}\mathcal{L}(\hat{Y}_T^{(i)}, Y_S^{(i)})$ After pruning the top-$\beta$ high-disagreement outliers, the remaining scores are partitioned into $k$ equipopulated strata. A budget of $m = \lfloor N(1 - \alpha) \rfloor$ examples is then allocated proportionally across the strata, sampling without replacement from the smallest stratum first to guarantee coverage of the gradient distribution. The procedure returns the transition set $\mathbb{S}_c$ that maximises teaching efficiency under the specified budget.

---

**Algorithm 1:** Quantifying gradient distribution and transition teaching set with coverage selection.

---

**Input:** $\mathcal{D}_S = \{(X_S^{(i)}, Y_S^{(i)}, \text{Index}_i)\}_{i=1}^N$ : $X_S^{(i)}$ is the input data; $Y_S^{(i)}$ is the label for example $i$; $\text{Index}_i$ represents the index of the example; $\mathcal{L}(\cdot, \cdot)$: loss function.

$\hat{Y}_T^{(i)}$: Output of the teacher model checkpoint for the $i$-th sample.

$\alpha$: dataset pruning rate; $\beta$: proportion of outliers to remove; $k$: the number of strata.

$K$: convergence checkpoints for score calculation.

**Step 1: Compute Information Quantity Scores** ;

**foreach** $i \in \{1, 2, \ldots, N\}$ **do**

    $\lfloor \mathcal{I}_{\text{Quant}}^{(i)} \leftarrow \nabla_{\theta_T^*}\mathcal{L}(\hat{Y}_T^{(i)}, Y_S^{(i)})^\top \nabla_{\theta_T^*}\mathcal{L}(\hat{Y}_T^{(i)}, Y_S^{(i)})$.

Update dataset: $\mathcal{D}_S \leftarrow \{(\mathcal{I}_{\text{Quant}}^{(i)}, \text{Index}_i)\}_{i=1}^N$.

**Step 2: Data Preprocessing**

1.Remove Outliers: Sort $\mathcal{D}_S$ by $\mathcal{I}_{\text{Quant}}^{(i)}$ and remove the largest outlier scoring $\lfloor n \cdot \beta \rfloor$ examples:

  $\mathcal{D}'_S \leftarrow \mathcal{D}_S \setminus \lfloor n \cdot \beta \rfloor$.

2.Divide the range of scores in $\mathcal{D}'_S$ into $k$ distinct strata: $R_1, R_2, \ldots, R_k$.

3.Initialize buckets: $\mathcal{Q} \leftarrow \{\mathbb{Q}_1, \mathbb{Q}_2, \ldots, \mathbb{Q}_k\}$, where each bucket $\mathbb{Q}_i$ contains examples with scores in range

  $R_i$.

4.Compute total budget for selected examples: $m \leftarrow \lfloor n \cdot (1 - \alpha) \rfloor$.

**Step 3: Select Examples from Strata**

Initialize transition set: $\mathbb{S}_c \leftarrow \varnothing$.

**while** $\mathcal{Q} \neq \varnothing$ **do**

    Identify the smallest bucket: $\mathbb{Q}_{\min} \leftarrow \underset{\mathbb{Q} \in \mathcal{Q}}{\arg\min} |\mathbb{Q}|$.

    Assign sampling quota: $m_Q \leftarrow \min\left\{|\mathbb{Q}_{\min}|, \left\lfloor \frac{m}{|\mathcal{Q}|} \right\rfloor\right\}$.

    $\mathbb{S}_Q \leftarrow$ randomly sample $m_Q$ examples from $\mathbb{Q}_{\min}$.

    Update the transition set: $\mathbb{S}_c \leftarrow \mathbb{S}_c \cup \mathbb{S}_Q$.

    Remove the processed stratum: $\mathcal{Q} \leftarrow \mathcal{Q} \setminus \{\mathbb{Q}_{\min}\}$.

    Adjust the remaining budget: $m \leftarrow m - m_Q$.

**Output:** Return the final transition set after preprocessing: $\mathbb{S}_c$.

---

# 6 Experiments

We typically assume that a well-established convergent teacher model and a known learner dataset, provides effective guidance, as it is trained on an IID dataset. However, ensuring that the learner's data distribution aligns with that of the teacher presents a challenge. Our analysis indicates that the mismatch in data distribution between the teacher and learner introduces teaching biases. In our experiments, we provide a case study in Figure 5 to validate this issue. To evaluate the effectiveness of our proposed transition machine teaching method in cases of data distribution mismatch, we configure the learner's dataset to include noisy or imbalanced data. Our experiments focus on image classification tasks, utilizing the MNIST (LeCun et al., 1998), CIFAR10 (Krizhevsky et al., 2009), CIFAR100 (Krizhevsky et al., 2009), and Tiny-ImageNet (Le & Yang, 2015) datasets, and employ four RTX A6000 GPUs for computation.

## 6.1 Setup

The experiment employs an optimizer based on stochastic gradient descent with Resnet18 (He et al., 2016), configured with a learning rate of 0.1, a momentum value of 0.9 to accelerate convergence, and a weight decay of 5e-4 to mitigate overfitting through regularization. To further enhance the training performance, we implement a cosine annealing learning rate scheduler, which dynamically adjusts the learning rate throughout the training duration according to a cosine curve.

The epochs for MNIST are set to 100 with a learning rate of 0.1, achieving a converged teacher test accuracy of approximately 99.50%. For CIFAR10, CIFAR100 and Tiny-ImageNet, the epochs are set to 200 with a learning rate of 0.1, resulting in converged test accuracies of about 95.17%, 78.11% and 61.01%. The teaching process utilizes a batch size of 100. We use 10,000 test images from the MNIST, CIFAR10, CIFAR100, and Tiny-ImageNet test datasets to evaluate the learners trained on the selected data. In analyzing the learner's data quantification information score, we use the following intermediate checkpoints from the teacher model: [1, 10, 20, 30, 60, 90, 120, 150, 180, 190, 192, 194, 196, 198, 200] for easier visualization of score variations. However, this is not a requirement for our method. In our method, we perform data quantization score calculations specifically at the converged checkpoint. For the core subset coverage selection process, we set the interval stratas to 50. IMT (Liu et al., 2017) and NIMT (Zhang et al., 2023) as our key baselines.

## 6.2 Teaching scenarios with noise disturbance

### 6.2.1 Comprehensive results under different data and noise ratios

Table 1: Test accuracies (%) of the learner trained with different data ratios (20%, 40%, 60%) and noise ratios (10%, 20%, 40%) across three datasets on UMT, IMT, NIMT and Transition MT.

| Dataset | Noise Ratio | Data Ratio | UMT | IMT | NIMT | Transition MT |
|---|---|---|---|---|---|---|
| MNIST | 10% | 20% | $97.93_{\pm 0.02}$ | $97.23_{\pm 0.03}$ | $98.47_{\pm 0.12}$ | $\mathbf{99.31_{\pm 0.01}}$ |
| | | 40% | $98.10_{\pm 0.07}$ | $97.40_{\pm 0.07}$ | $98.63_{\pm 0.19}$ | $\mathbf{99.49_{\pm 0.02}}$ |
| | | 60% | $98.42_{\pm 0.02}$ | $98.31_{\pm 0.06}$ | $98.45_{\pm 0.25}$ | $\mathbf{99.56_{\pm 0.01}}$ |
| | 20% | 20% | $95.08_{\pm 0.03}$ | $90.54_{\pm 0.18}$ | $96.94_{\pm 0.31}$ | $\mathbf{99.32_{\pm 0.07}}$ |
| | | 40% | $96.41_{\pm 0.04}$ | $95.61_{\pm 0.10}$ | $97.27_{\pm 0.31}$ | $\mathbf{99.54_{\pm 0.01}}$ |
| | | 60% | $96.73_{\pm 0.03}$ | $96.45_{\pm 0.09}$ | $97.01_{\pm 0.25}$ | $\mathbf{99.68_{\pm 0.01}}$ |
| | 40% | 20% | $82.12_{\pm 0.09}$ | $81.62_{\pm 0.09}$ | $86.43_{\pm 0.14}$ | $\mathbf{99.32_{\pm 0.02}}$ |
| | | 40% | $83.04_{\pm 0.07}$ | $82.53_{\pm 0.09}$ | $87.64_{\pm 0.17}$ | $\mathbf{99.59_{\pm 0.01}}$ |
| | | 60% | $83.45_{\pm 0.06}$ | $83.60_{\pm 0.07}$ | $86.06_{\pm 0.15}$ | $\mathbf{99.54_{\pm 0.01}}$ |
| CIFAR10 | 10% | 20% | $81.55_{\pm 0.12}$ | $81.86_{\pm 0.12}$ | $81.96_{\pm 0.15}$ | $\mathbf{86.64_{\pm 0.17}}$ |
| | | 40% | $85.80_{\pm 0.14}$ | $86.47_{\pm 0.10}$ | $87.55_{\pm 0.12}$ | $\mathbf{91.59_{\pm 0.12}}$ |
| | | 60% | $88.19_{\pm 0.22}$ | $88.25_{\pm 0.15}$ | $89.58_{\pm 0.05}$ | $\mathbf{93.14_{\pm 0.09}}$ |
| | 20% | 20% | $74.58_{\pm 0.16}$ | $74.75_{\pm 0.24}$ | $75.01_{\pm 0.11}$ | $\mathbf{87.51_{\pm 0.11}}$ |
| | | 40% | $79.73_{\pm 0.18}$ | $80.13_{\pm 0.17}$ | $81.73_{\pm 0.14}$ | $\mathbf{91.91_{\pm 0.10}}$ |
| | | 60% | $81.70_{\pm 0.16}$ | $81.69_{\pm 0.17}$ | $84.34_{\pm 0.04}$ | $\mathbf{93.45_{\pm 0.08}}$ |
| | 40% | 20% | $54.59_{\pm 0.16}$ | $54.77_{\pm 0.16}$ | $55.97_{\pm 0.16}$ | $\mathbf{87.53_{\pm 0.13}}$ |
| | | 40% | $62.94_{\pm 0.12}$ | $62.57_{\pm 0.37}$ | $63.61_{\pm 0.19}$ | $\mathbf{91.75_{\pm 0.07}}$ |
| | | 60% | $64.58_{\pm 0.28}$ | $63.19_{\pm 0.28}$ | $66.83_{\pm 0.28}$ | $\mathbf{93.49_{\pm 0.14}}$ |
| CIFAR100 | 10% | 20% | $50.245_{\pm 0.17}$ | $49.43_{\pm 0.18}$ | $49.16_{\pm 0.18}$ | $\mathbf{56.39_{\pm 0.16}}$ |
| | | 40% | $61.525_{\pm 0.09}$ | $61.43_{\pm 0.11}$ | $60.65_{\pm 0.18}$ | $\mathbf{68.69_{\pm 0.17}}$ |
| | | 60% | $65.605_{\pm 0.13}$ | $66.12_{\pm 0.17}$ | $64.45_{\pm 0.21}$ | $\mathbf{73.09_{\pm 0.09}}$ |
| | 20% | 20% | $42.754_{\pm 0.14}$ | $41.79_{\pm 0.16}$ | $43.06_{\pm 0.14}$ | $\mathbf{56.68_{\pm 0.23}}$ |
| | | 40% | $55.024_{\pm 0.17}$ | $55.25_{\pm 0.15}$ | $54.25_{\pm 0.11}$ | $\mathbf{68.20_{\pm 0.12}}$ |
| | | 60% | $59.493_{\pm 0.17}$ | $59.80_{\pm 0.12}$ | $58.24_{\pm 0.14}$ | $\mathbf{72.75_{\pm 0.16}}$ |
| | 40% | 20% | $27.31_{\pm 0.16}$ | $25.69_{\pm 0.16}$ | $27.35_{\pm 0.18}$ | $\mathbf{56.21_{\pm 0.18}}$ |
| | | 40% | $40.94_{\pm 0.16}$ | $40.53_{\pm 0.17}$ | $40.53_{\pm 0.19}$ | $\mathbf{67.62_{\pm 0.14}}$ |
| | | 60% | $45.15_{\pm 0.16}$ | $45.73_{\pm 0.18}$ | $45.43_{\pm 0.16}$ | $\mathbf{73.24_{\pm 0.17}}$ |

Table 1 illustrates the analysis of test accuracies for four kinds of methods on three datasets under different noise rates and data proportions, and Uniform MT (UMT) refers to random teaching without strategy. We inject noise by flipping labels, with flip rates set to 10%, 20%, and 40% to simulate noisy datasets. In addition, we vary the proportions of training data (20%, 40%, and 60%) to examine how different levels of available data influence the effectiveness of each method. Additionally, we randomly plot the distribution of the selected learners' teaching influence quantization scores in Appenddix B Figure 6. It is clear from the figure that the noisy data has been effectively pruned.

**Results.** Specifically, on the MNIST dataset, our method achieves the highest performance across all settings, with accuracy ranging from 99.31% to 99.68%, even under severe noise disturbance. For CIFAR10, our method consistently outperforms both baselines and the uniform strategy across various noise rates and data ratios, reaching up to 93.49%. On CIFAR100, despite the baselines suffering from notable degradation due to increased category difficulty and noise sensitivity, Transition MT still delivers significant improvements, particularly under high noise conditions, with gains exceeding 25% absolute accuracy in some cases. These results suggest that Transition MT effectively mitigates the adverse impact of noisy labels by leveraging gradient-based disagreement detection, thereby reducing teaching bias introduced by distribution mismatch. Moreover, as the data ratio increases, all methods show improved accuracy, but Transition MT exhibits the most pronounced improvements, especially on the more challenging CIFAR datasets.

### 6.2.2 Scalability and generalization on Tiny-ImageNet

Table 2: Test Accuracy (%) on Tiny-ImageNet under Different Data Ratios and Noise Levels ("N" in the table represents noise level).

| Method | Data Ratio = 0.2 | | | Data Ratio = 0.4 | | | Data Ratio = 0.6 | | |
|---|---|---|---|---|---|---|---|---|---|
| | N=0.1 | N=0.2 | N=0.4 | N=0.1 | N=0.2 | N=0.4 | N=0.1 | N=0.2 | N=0.4 |
| UMT | 32.46 | 27.65 | 16.05 | 44.94 | 37.68 | 25.32 | 50.72 | 45.59 | 31.81 |
| NIMT | 32.96 | 26.75 | 13.83 | 45.35 | 39.71 | 25.96 | 51.20 | 45.74 | 33.01 |
| **Transition MT (Ours)** | **38.35** | **36.62** | **36.08** | **50.46** | **50.15** | **50.06** | **56.03** | **55.97** | **55.90** |

To further examine the scalability of Transition MT on larger and more complex datasets, we extend the experiments to Tiny-ImageNet. This dataset contains 200 classes with 500 training images and 50 validation images per class, thus providing a more challenging teaching scenario with higher intra-class diversity and limited per-class samples. We adopt the same setup as in CIFAR experiments, introducing varying noise ratios (10%, 20%, 40%) and selected data proportions (20%, 40%, 60%).

**Results.** As shown in Table 2, both UMT and NIMT experience substantial accuracy drops when data ratios are low or noise levels are high. In contrast, Transition MT consistently delivers superior results, achieving up to 56.03% accuracy, a considerable margin above the baselines. Importantly, the gap between Transition MT and the baselines widens as the noise level increases, highlighting its robustness to noisy teaching conditions. These findings confirm that Transition MT generalizes effectively, maintaining strong performance on large and challenging benchmarks.

### 6.2.3 Comparative experiments with other existing methods

To empirically evaluate the effectiveness of the proposed Transition MT method, we compare it against the strong subset selection baseline Probabilistic Bilevel Coreset Selection (PBCS) (Zhou et al., 2022) on CIFAR10. While both approaches aim to improve learning under limited or noisy data, they are grounded in fundamentally different paradigms: Transition MT exploits the knowledge of a pre-trained teacher model to adaptively guide the learner, whereas PBCS formulates coreset selection as a continuous probabilistic bilevel optimization problem, assigning and refining sample weights in a purely data-centric manner. This comparison thus provides a critical test of whether task-specific teaching strategies can surpass general-purpose subset selection methods in challenging learning scenarios.

Table 3: Comparison of Test Accuracy (%) on CIFAR10 between PBCS (Zhou et al., 2022) and Transition MT (Ours) under Different Data Ratios and Noise Levels ("N" in the table represents noise level).

| Method | Data Ratio = 0.2 | | | Data Ratio = 0.4 | | | Data Ratio = 0.6 | | |
|---|---|---|---|---|---|---|---|---|---|
| | N=0.1 | N=0.2 | N=0.4 | N=0.1 | N=0.2 | N=0.4 | N=0.1 | N=0.2 | N=0.4 |
| PBCS | 81.97 ±0.13 | 72.16 ±0.17 | 56.94 ±0.20 | 86.20 ±0.15 | 80.25 ±0.16 | 62.06 ±0.20 | 88.17 ±0.09 | 81.91 ±0.14 | 63.57 ±0.24 |
| **Transition MT (Ours)** | **86.64** ±0.17 | **86.51** ±0.11 | **86.53** ±0.13 | **91.59** ±0.12 | **91.91** ±0.10 | **91.75** ±0.07 | **93.14** ±0.09 | **93.45** ±0.08 | **93.49** ±0.14 |

**Results.** Table 3 reports the results. Transition MT demonstrates superior or highly competitive performance across all data ratios and noise levels. While PBCS suffers substantial performance degradation under high noise (e.g., dropping to 56.94% at 20% data ratio and 40% noise), Transition MT maintains remarkably high accuracy. This suggests that explicitly modeling teacher–learner dynamics is more effective in noise-robust teaching scenarios than data-only subset selection. Overall, the results highlight the unique advantage of Transition MT as a dedicated teaching algorithm, outperforming the coreset selection methods by large margins in high-noise conditions while remaining competitive in less noisy settings.

Overall, Transition Machine Teaching consistently achieves strong results across datasets, data ratios, and noise levels. Its advantages are particularly evident under challenging conditions, such as high noise or complex datasets, where traditional baselines and even advanced coreset selection methods exhibit significant degradation. These results provide solid empirical evidence that Transition MT effectively addresses the fundamental issue of distribution mismatch and teaching bias, validating its role as a principled and robust machine teaching framework.

### 6.3 Teaching scenarios with unbalanced distribution

Table 4: Test accuracies (%) of different methods under varying selected data ratios (20%, 40%, 60%, 80%) on MNIST, CIFAR10, and CIFAR100 under the teaching scenarios with unbalanced distribution.

| Method | MNIST | | | | CIFAR10 | | | | CIFAR100 | | | |
|---|---|---|---|---|---|---|---|---|---|---|---|---|
| | 20% | 40% | 60% | 80% | 20% | 40% | 60% | 80% | 20% | 40% | 60% | 80% |
| IMT | 98.39 ±0.06 | 98.79 ±0.07 | 98.93 ±0.09 | 99.00 ±0.08 | 73.53 ±0.31 | 84.69 ±0.30 | 84.71 ±0.18 | 86.10 ±0.23 | 38.20 ±0.17 | 52.39 ±0.12 | 53.95 ±0.15 | 62.87 ±0.08 |
| NIMT | 98.36 ±0.04 | 98.84 ±0.07 | 98.86 ±0.09 | 98.81 ±0.07 | 70.04 ±0.21 | 82.06 ±0.15 | 84.92 ±0.16 | 86.66 ±0.17 | 38.65 ±0.15 | 52.23 ±0.16 | 56.71 ±0.16 | 62.43 ±0.13 |
| UMT | 94.77 ±0.10 | 97.82 ±0.07 | 98.32 ±0.07 | 98.70 ±0.10 | 74.61 ±0.12 | 80.78 ±0.17 | 83.89 ±0.18 | 85.12 ±0.13 | 37.80 ±0.11 | 52.73 ±0.16 | 58.59 ±0.14 | 63.23 ±0.12 |
| Transition MT | **98.78** ±0.03 | **98.91** ±0.04 | **99.11** ±0.03 | **99.33** ±0.02 | **80.67** ±0.44 | **84.89** ±0.18 | **86.50** ±0.23 | **86.79** ±0.19 | **40.95** ±0.14 | **56.65** ±0.24 | **62.11** ±0.10 | **64.41** ±0.13 |

In this section, we use a Dirichlet distribution to randomly initialize class quantities for simulating imbalanced datasets. For the concentration parameter $\alpha$ of the Dirichlet distribution, we randomly set it to [0.4, 1.1, 0.6, 0.1, 0.9, 1.3, 1.4, 1.2, 0.5, 0.7] for MNIST, and [0.4, 1.1, 0.6, 0.5, 0.9, 1.1, 1.5, 1.2, 0.5, 0.7] for CIFAR10. For CIFAR100, due to the larger number of categories, we randomly generated values using a normal distribution with a mean of 1.0 and a variance of 0.5. For CIFAR10 and CIFAR100 datasets, we construct an imbalanced dataset consisting of 30,000 data points. For the MNIST dataset, we construct an imbalanced dataset consisting of 40,000 data points. We test the learner's accuracy with learner data subsets comprising 20%, 40%, 60%, and 80% of the total data. Our results, shown in the Table 4, indicate a significant improvement. In Appendix B, Figure 7, we present an illustration of class imbalance between CIFAR10 and CIFAR100 .

**Results.** In all datasets (MNIST, CIFAR10, CIFAR100) and under all data ratio settings, Transition MT consistently outperforms IMT and NIMT. The leading advantage is especially pronounced at lower data ratios. As the data ratio increases, the accuracy of all methods improves, but Transition MT shows a faster and greater increase, such as from 40.95% to 64.41% on CIFAR100, an improvement of about 23.5%. This

indicates that CIFAR100's performance is more sensitive to changes in sample size. On CIFAR10, the improvement of Transition MT tends to saturate, suggesting that the gradient disagreement samples have been effectively pruned away.

**Teaching influence score analysis.** In imbalanced teaching scenarios, we calculate the frequency distribution of teaching influence scores for data points, as shown in Figure 4. As the model converges, the frequency distribution of score value shows an overall trend of evolving from a relatively dispersed distribution to being highly concentrated in the low score value interval. At this stage, samples in the tail are typically gradient-disagreement data samples, which we avoid during the teaching process. Similarly, we plot the score distribution using data indices as the variable in Appendix B Figure 8. As the model converges, most data samples' scores converge to a fixed range, with only a few gradient-disagreement samples remaining as outliers. This observation validates our selection strategy in Eq. 8, which excludes gradient-disagreement samples from the learning sequence.

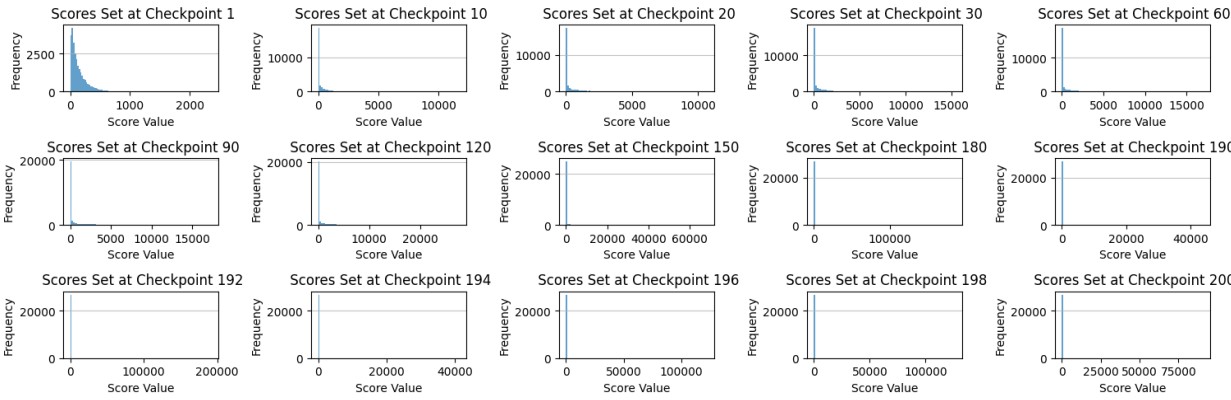

Figure 4: Score frequency histogram, teaching influence score is obtained using 40% of the sample size on the imbalanced dataset of CIFAR10. It is worth noting that the teaching influence scores will converge to a very small range after convergence.

# 7 Conclusion

In summary, we examine the iterative interactions between the teacher and learner in machine teaching, revealing cumulative teaching bias caused by data distribution mismatches. To address this challenge, we propose a novel method that surpasses the limitations of conventional approaches in handling model misalignment and data distribution disparities. By transferring the learner's data information to the gradient space of the teacher model, we effectively infer quantization scores for the learner's data within the trust region, optimizing the selection of learner samples. This strategy requires only a single forward and backward pass, thereby avoiding the cumulative biases associated with multiple feedback interactions while maintaining a balance between informativeness and representativeness. Rigorous theoretical analysis and extensive experiments validate its effectiveness. In the future, we will focus on developing more precise methods for higher-order information and data interactions within the model.

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

# A  Appendix A

## A.1  The Proof of Proposition 4.1

**Proposition A.1.** *In iterative teaching process, the learner's candidate dataset is $\mathcal{D}_S \sim \mathcal{P}$, and the teacher's goal is to enable the learner to learn a model that is functionally close to the $\theta_T^*$. For the learner, teacher dataset $\mathcal{D}_T \sim \mathcal{T}$ is typically a black box. The selection process for each batch of samples $(x, y)$ follows the procedure in Eq. 2 (or Eq. 3).* **Thus, we can conclude that in nonlinear composite structures such as neural networks, when there is a distribution mismatch ($\mathcal{P} \neq \mathcal{T}$), the gradient $\nabla_\theta \mathcal{L}(f_{\hat{\theta}_S^t}(x), y)$ on $\mathcal{P}$ can lead to an update direction for the learner model that deviates from $\theta_T^*$. Based on LMC theory, the limitations of the high-dimensional metric $\|\cdot\|_2^2$ are further exacerbated in cases of distribution mismatch, amplifying the sample selection bias.**

*Proof.* We reveal the limitations of Euclidean distance in parameter space by analyzing the phenomenon of LMC in neural networks and illustrate the impact of distribution mismatch on sample selection. We utilize the LMC phenomenon to highlight the intrinsic biases of linear operations in neural networks. In LMC, let $\theta_1$ and $\theta_2$ represent two model parameters with different initial values but trained on the same dataset in differing orders, ultimately converging to models with similar performance. When no adjustments are made to $\theta_1$ and $\theta_2$, the loss function $\mathcal{L}$ of neural network $f$ satisfies the following inequality:

$$\mathcal{L}(f_{\lambda\theta_1 + (1-\lambda\theta_2)}) = \mathcal{L}(f_{\lambda(\theta_1 - \theta_2) + \theta_2}) > \mathcal{L}(f_{\theta_1}) \approx \mathcal{L}(f_{\theta_2}).$$

This indicates that $\theta_1$ and $\theta_2$ are not directly "connected" in parameter space, the Euclidean linear interpolation path passes through high-loss regions, a consequence of the highly non-convex nature of neural network space. This suggests that directly using the Euclidean distance $\|\theta_1 - \theta_2\|_2^2$ in IMT cannot effectively connect the parameter spaces of two networks, thus failing to reflect functional equivalence. By applying a permutation transformation $\pi = \{P_1, \cdots, P_{L-1}\}$ to $\theta_1$ as described in (Ainsworth et al., 2023), we can obtain $\pi(\theta_1)$ such that the linear interpolation satisfies:

$$\mathcal{L}(f_{\lambda(\pi(\theta_1) - \theta_2) + \theta_2}) = \mathcal{L}(f_{\lambda\pi(\theta_1) - \theta_2(\lambda-1)}) \approx \mathcal{L}(f_{\theta_1}) \approx \mathcal{L}(f_{\theta_2})$$

It indicates that permutation transformations can align functional equivalence points in the parameter space, reducing the loss barriers in linear interpolation. Therefore, applying permutation transformations to the learner's parameters $\hat{\theta}_S^{t+1}$ in iterative machine teaching helps mitigate functional errors caused by misaligned parameters. However, when $\mathcal{P} \neq \mathcal{T}$, the learner's gradient update $\hat{\theta}_S^{t+1} = \hat{\theta}_S^t - \eta_t \nabla_\theta \mathcal{L}(f_{\hat{\theta}_S^t}(x), y)$ is based on $\mathcal{P}$, while the target $\theta_T^*$ is optimized based on $\mathcal{T}$. The distribution discrepancy leads the gradient direction to deviate from $\theta_T^*$, causing errors in $\left\|\hat{\theta}_S^{t+1} - \theta_T^*\right\|_2^2$ and making the sample selection $d_{t+1}$ potentially suboptimal. This accumulation of bias can slow the learner's convergence and result in $\hat{\theta}_S^{t+1}$ deviating from the functionally equivalent target. $\square$

## A.2  The proof of Eq. 5

We have demonstrated the generalization error between the learner's actual data distribution $\mathcal{Q}$ and the original data distribution $\mathcal{P}$. The error of the actual distribution is often larger, while under ideal conditions, the learner's original distribution can be considered approximately equal to the teacher's data distribution $\mathcal{T}$. Therefore, it is evident that there is a significant generalization disagreement between the learner's actual data distribution and the teacher's data distribution.

**Lemma A.1.** *Upper Bound on Model Generalization Performance under Learner's Shifted Distribution: Let $\mathcal{L} : Y \times Y \to \mathbf{R}^+$ be the loss function. For some $M > 0$, assume $\sup_{(x,y)\in(X,Y)} |\mathcal{L}(h_\theta(x), y)| \leq M$. Then for any probability measure $\mathcal{P}$ and $\epsilon > 0$, there exists an expected inequality for the learner's data distribution $\mathcal{P}$ and the learner's shifted data distribution $\mathcal{Q}$:*

$$\sup_{\mathcal{Q} \in B_\epsilon(\mathcal{P})} \mathbb{E}_{\mathcal{Q}}[\mathcal{L}(h_\theta(x), y)] \leq \mathbb{E}_{\mathcal{P}}[\mathcal{L}(h_\theta(x), y)]$$

$$+2\lambda_\epsilon \left[\mathbb{V}_{\mathcal{P}}[\mathcal{L}(h_\theta(x), y)]\right]^{1/2} + \Delta_\epsilon^u(\epsilon, M, \mathbb{E}_{\mathcal{P}}, \mathbb{V}_{\mathcal{P}}),$$

*where*

$$\Delta_{\epsilon}^{u}(\epsilon, M, \mathbb{E}_{\mathcal{P}}, \mathbb{V}_{\mathcal{P}}) = \epsilon^2 \left(2 - \epsilon^2\right) \left[M - \mathbb{E}_{\mathcal{P}}[\mathcal{L}(h_{\theta}(x), y)] - \frac{\mathbb{V}_{\mathcal{P}}[\mathcal{L}(h_{\theta}(x), y)]}{M - \mathbb{E}_{\mathcal{P}}[\mathcal{L}(h_{\theta}(x), y)]}\right],$$

*and $\mathbb{E}$ denotes expectation, $\mathbb{V}$ denotes variance, and $\lambda_{\epsilon} = \left[\epsilon^2 \left(2 - \epsilon^2\right) \left(1 - \epsilon^2\right)^2\right]^{1/2}$. The set $B_{\epsilon}(\mathcal{P}) = \{\mathcal{Q} \in \mathcal{P}(X, Y) : H(\mathcal{P}, \mathcal{Q}) \leq \epsilon\}$ is the Hellinger ball centered at $\mathcal{P}$ with radius $\epsilon$, such that the radius $\epsilon$ satisfies:*

$$\epsilon^2 \leq 1 - \left[1 + \frac{(M - \mathbb{E}_{\mathcal{P}}[\mathcal{L}(h_{\theta}(x), y)])^2}{\mathbb{V}_{\mathcal{P}}[\mathcal{L}(h_{\theta}(x), y)]}\right]^{-1/2}.$$

*Proof.* **The expected value and variance induced by the properties of the Gram determinant.** From the structure of the Gram matrix $G_f$:

$$G_f := \begin{bmatrix} \psi_{\mathcal{Q}} \\ \psi_{\mathcal{P}} \\ f\psi_{\mathcal{P}} \end{bmatrix} \begin{bmatrix} \psi_{\mathcal{Q}} & \psi_{\mathcal{P}} & f\psi_{\mathcal{P}} \end{bmatrix} = \begin{bmatrix} 1 & \langle \psi_{\mathcal{Q}}, \psi_{\mathcal{P}} \rangle & \langle \psi_{\mathcal{Q}}, f\psi_{\mathcal{P}} \rangle \\ \langle \psi_{\mathcal{P}}, \psi_{\mathcal{Q}} \rangle & 1 & \langle \psi_{\mathcal{P}}, f\psi_{\mathcal{P}} \rangle \\ \langle f\psi_{\mathcal{P}}, \psi_{\mathcal{Q}} \rangle & \langle f\psi_{\mathcal{P}}, \psi_{\mathcal{P}} \rangle & \langle f\psi_{\mathcal{P}}, f\psi_{\mathcal{P}} \rangle \end{bmatrix},$$

where $\psi_{\mathcal{P}}$ and $\psi_{\mathcal{Q}}$ represent functions derived from data distributions $\mathcal{P}$ and $\mathcal{Q}$. The term $f$ is a function that transforms $\psi_{\mathcal{P}}$. We can compute the determinant of the matrix $G_f$:

$$\begin{aligned} |G_f| &= \langle f\psi_{\mathcal{P}}, f\psi_{\mathcal{P}} \rangle - \langle f\psi_{\mathcal{P}}, \psi_{\mathcal{P}} \rangle^2 - \langle \psi_{\mathcal{Q}}, \psi_{\mathcal{P}} \rangle \left(\langle \psi_{\mathcal{Q}}, \psi_{\mathcal{P}} \rangle \langle f\psi_{\mathcal{P}}, f\psi_{\mathcal{P}} \rangle - \langle \psi_{\mathcal{Q}}, f\psi_{\mathcal{P}} \rangle \langle f\psi_{\mathcal{P}}, \psi_{\mathcal{P}} \rangle\right) \\ &\quad + \langle f\psi_{\mathcal{P}}, \psi_{\mathcal{Q}} \rangle \left(\langle \psi_{\mathcal{P}}, \psi_{\mathcal{Q}} \rangle \langle \psi_{\mathcal{P}}, f\psi_{\mathcal{P}} \rangle - \langle f\psi_{\mathcal{P}}, \psi_{\mathcal{Q}} \rangle\right) \\ &= \langle f\psi_{\mathcal{P}}, f\psi_{\mathcal{P}} \rangle - \langle f\psi_{\mathcal{P}}, \psi_{\mathcal{P}} \rangle^2 - \langle \psi_{\mathcal{Q}}, \psi_{\mathcal{P}} \rangle^2 \langle f\psi_{\mathcal{P}}, f\psi_{\mathcal{P}} \rangle + \langle \psi_{\mathcal{Q}}, \psi_{\mathcal{P}} \rangle \langle \psi_{\mathcal{Q}}, f\psi_{\mathcal{P}} \rangle \langle f\psi_{\mathcal{P}}, \psi_{\mathcal{P}} \rangle \\ &\quad + \langle f\psi_{\mathcal{P}}, \psi_{\mathcal{Q}} \rangle \langle \psi_{\mathcal{P}}, \psi_{\mathcal{Q}} \rangle \langle \psi_{\mathcal{P}}, f\psi_{\mathcal{P}} \rangle - \langle f\psi_{\mathcal{P}}, \psi_{\mathcal{Q}} \rangle^2 \\ &= -\langle f\psi_{\mathcal{P}}, \psi_{\mathcal{Q}} \rangle^2 + 2\langle \psi_{\mathcal{P}}, \psi_{\mathcal{Q}} \rangle \langle f\psi_{\mathcal{P}}, \psi_{\mathcal{P}} \rangle \langle f\psi_{\mathcal{P}}, \psi_{\mathcal{Q}} \rangle + \left(1 - \langle \psi_{\mathcal{Q}}, \psi_{\mathcal{P}} \rangle^2\right) \langle f\psi_{\mathcal{P}}, f\psi_{\mathcal{P}} \rangle - \langle f\psi_{\mathcal{P}}, \psi_{\mathcal{P}} \rangle^2. \end{aligned}$$

Let us define: $x = \langle f\psi_{\mathcal{P}}, \psi_{\mathcal{Q}} \rangle$, $b = 2\langle \psi_{\mathcal{P}}, \psi_{\mathcal{Q}} \rangle \langle f\psi_{\mathcal{P}}, \psi_{\mathcal{P}} \rangle$, $c = \left(1 - \langle \psi_{\mathcal{Q}}, \psi_{\mathcal{P}} \rangle^2\right) \langle f\psi_{\mathcal{P}}, f\psi_{\mathcal{P}} \rangle - \langle f\psi_{\mathcal{P}}, \psi_{\mathcal{P}} \rangle^2$. Thus, we express the determinant as:

$$|G_f| = -x^2 + bx + c.$$

The property of the Gram matrix $\det(G_f) \geq 0$ implies that:

$$\frac{b}{2} - \sqrt{\frac{b^2}{4} + c} \leq x = \langle f\psi_{\mathcal{P}}, \psi_{\mathcal{Q}} \rangle \leq \frac{b}{2} + \sqrt{\frac{b^2}{4} + c}, \qquad (1)$$

where

$$\frac{b}{2} = \langle \psi_{\mathcal{P}}, \psi_{\mathcal{Q}} \rangle \langle f\psi_{\mathcal{P}}, \psi_{\mathcal{P}} \rangle,$$

$$\sqrt{\frac{b^2}{4} + c} = \sqrt{\langle f\psi_{\mathcal{P}}, \psi_{\mathcal{Q}} \rangle^2 \langle \psi_{\mathcal{P}}, \psi_{\mathcal{P}} \rangle^2 + (1 - \langle \psi_{\mathcal{Q}}, \psi_{\mathcal{P}} \rangle^2) \langle f\psi_{\mathcal{P}}, f\psi_{\mathcal{P}} \rangle - \langle f\psi_{\mathcal{P}}, \psi_{\mathcal{P}} \rangle^2}.$$

To simplify, let:

$$\Delta F^2 = \langle f\psi_{\mathcal{P}}, f\psi_{\mathcal{P}} \rangle - \langle f\psi_{\mathcal{P}}, \psi_{\mathcal{P}} \rangle^2, \quad S = \langle \psi_{\mathcal{Q}}, \psi_{\mathcal{P}} \rangle.$$

Then, we have: $\sqrt{\frac{b^2}{4} + c} = \sqrt{(1 - S^2)\Delta F^2}$. Consequently, we obtain the inequality:

$$\langle f\psi_{\mathcal{P}}, \psi_{\mathcal{Q}} \rangle \geq S\langle f\psi_{\mathcal{P}}, \psi_{\mathcal{P}} \rangle - \sqrt{1 - S^2}\Delta F. \qquad (2)$$

By the Cauchy-Schwarz inequality, we have:

$$\langle f\psi_{\mathcal{P}}, \psi_{\mathcal{Q}}\rangle \le \sqrt{\langle \psi_{\mathcal{Q}}, f\psi_{\mathcal{Q}}\rangle}\sqrt{\langle \psi_{\mathcal{P}}, f\psi_{\mathcal{P}}\rangle} \;\Rightarrow\; \langle \psi_{\mathcal{Q}}, f\psi_{\mathcal{Q}}\rangle \ge \frac{\langle f\psi_{\mathcal{P}}, \psi_{\mathcal{Q}}\rangle^2}{\langle f\psi_{\mathcal{P}}, \psi_{\mathcal{P}}\rangle}.$$

Finally, according to (2), we deduce:

$$\langle \psi_{\mathcal{Q}}, f\psi_{\mathcal{Q}}\rangle \ge \frac{\left\{S\langle f\psi_{\mathcal{P}}, \psi_{\mathcal{P}}\rangle - \sqrt{1-S^2}\Delta F\right\}^2}{\langle \psi_{\mathcal{P}}, f\psi_{\mathcal{P}}\rangle}.$$

Additionally, we can express:

$$\langle \psi_{\mathcal{Q}}, f\psi_{\mathcal{Q}}\rangle \ge S^2\langle f\psi_{\mathcal{P}}, \psi_{\mathcal{P}}\rangle - 2S\sqrt{1-S^2}\langle f\psi_{\mathcal{P}}, \psi_{\mathcal{P}}\rangle\Delta F + \frac{(1-S^2)\Delta F^2}{\langle \psi_{\mathcal{P}}, f\psi_{\mathcal{P}}\rangle}. \quad (3)$$

We define the variables as follows: $\psi_{\mathcal{P}} := \sqrt{\frac{d\mathcal{P}}{d\mu}}, \psi_{\mathcal{Q}} := \sqrt{\frac{d\mathcal{Q}}{d\mu}}$. The inner product in $\mathcal{L}_2$ space is defined as $\langle f, g\rangle_{\mathcal{L}_2} = \int_Z fg\, d\mu$. The operator $M_f$ is defined by $(M_f\psi)(z) = f(z)\cdot\psi(z)$. Thus, we conclude that:

$$\mathbb{E}_{z\sim\mathcal{P}}[f(z)] = \int_Z f(z)\, d\mathcal{P}(z) = \int_Z f(z)\frac{d\mathcal{P}}{d\mu}(z)\, d\mu(z) = \langle \psi_{\mathcal{P}}, M_f\psi_{\mathcal{P}}\rangle_{\mathcal{L}_2}.$$

Similarly, the variance of $f(Z)$ is given by:

$$\mathbb{V}_{z\sim\mathcal{P}}[f(z)] = \langle \psi_{\mathcal{P}}, M_f\psi_{\mathcal{P}}\rangle_{\mathcal{L}_2} - \langle \psi_{\mathcal{P}}, M_f\psi_{\mathcal{P}}\rangle_{\mathcal{L}_2}^2.$$

**Upper bound of $\mathbb{E}_{\mathcal{Q}}[\mathcal{L}(\mathcal{Z})]$.** Let $\mathcal{Z} = (X, Y)$, for $\sup_{z\in\mathcal{Z}}|\mathcal{L}(z)| \le M$, it follows that the function $f_{\mathcal{L}}(\cdot) = M - \mathcal{L}(\cdot)$ is essentially bounded with respect to $\mu$ and hence difines a bounded linear operator. Substitute the items in inequality (3), we get the bound:

$$\mathbb{E}_{\mathcal{Q}}[M - \mathcal{L}(\mathcal{Z})] \ge |\langle \psi_{\mathcal{P}}, \psi_{\mathcal{Q}}\rangle|^2\, \mathbb{E}_P[M - \mathcal{L}(\mathcal{Z})] - 2\,|\langle \psi_{\mathcal{P}}, \psi_{\mathcal{Q}}\rangle|\sqrt{\left(1 - |\langle \psi_{\mathcal{P}}, \psi_{\mathcal{Q}}\rangle|^2\right)\mathbb{V}_{\mathcal{P}}[M - \mathcal{L}(\mathcal{Z})]}$$

$$+ \frac{\left(1 - |\langle \psi_{\mathcal{P}}, \psi_{\mathcal{P}}\rangle|^2\right)\mathbb{V}_{\mathcal{P}}[M - \mathcal{L}(\mathcal{Z})]}{\mathbb{E}_{\mathcal{P}}[M - \mathcal{L}(\mathcal{Z})]},$$

and

$$\mathbb{E}_{\mathcal{Q}}[\mathcal{L}(\mathcal{Z})] \le |\langle \psi_{\mathcal{P}}, \psi_{\mathcal{Q}}\rangle|^2\, \mathbb{E}_{\mathcal{P}}[\mathcal{L}(\mathcal{Z}) - M] + M + 2\,|\langle \psi_{\mathcal{P}}, \psi_{\mathcal{Q}}\rangle|\sqrt{\left(1 - |\langle \psi_{\mathcal{P}}, \psi_{\mathcal{Q}}\rangle|^2\right)\mathbb{V}_{\mathcal{P}}[M - \mathcal{L}(\mathcal{Z})]}$$

$$- \frac{\left(1 - |\langle \psi_{\mathcal{P}}, \psi_{\mathcal{P}}\rangle|^2\right)\mathbb{V}_{\mathcal{P}}[M - \mathcal{L}(\mathcal{Z})]}{\mathbb{E}_{\mathcal{P}}[M - \mathcal{L}(\mathcal{Z})]}.$$

For

$$\mathbb{V}_{\mathcal{P}}[M - \mathcal{L}(\mathcal{Z})] = \mathbb{V}_{\mathcal{P}}[\mathcal{L}(\mathcal{Z})],\; \langle \psi_{\mathcal{P}}, \psi_{\mathcal{Q}}\rangle = 1 - H^2(\mathcal{P}, \mathcal{Q}) = 1 - \epsilon^2,$$

$$H^2(\mathcal{P}, \mathcal{Q}) = \frac{1}{2}\int_Z (\psi_{\mathcal{P}} - \psi_{\mathcal{Q}})^2\, d\mu = 1 - \int_Z \psi_{\mathcal{P}}\psi_{\mathcal{Q}}d\mu = 1 - \langle \psi_{\mathcal{P}}, \psi_{\mathcal{Q}}\rangle,$$

$$\left(1 - \left(1 - \epsilon^2\right)^2\right)\mathbb{E}_{\mathcal{P}}\left[\mathcal{L}(\mathcal{Z}) - M\right] = \epsilon^2\left(2 - \epsilon^2\right)\mathbb{E}_{\mathcal{P}}[\mathcal{L}(\mathcal{Z}) - M],$$

$$\left(\left(1 - \epsilon^2\right)^2 - 1\right)\mathbb{E}_{\mathcal{P}}[\mathcal{L}(\mathcal{Z}) - M] = \left(1 - \epsilon^2\right)^2\mathbb{E}_{\mathcal{P}}\left[\mathcal{L}(\mathcal{Z}) - M\right] - \mathbb{E}_{\mathcal{P}}[\mathcal{L}(\mathcal{Z}) - M],$$

$$2\left(1 - \epsilon^2\right)\sqrt{\left(1 - (1 - \epsilon^2)^2\right)\mathbb{V}_{\mathcal{P}}[\mathcal{L}(\mathcal{Z})]} = 2\sqrt{\epsilon^2\left(2 - \epsilon^2\right)\left(1 - \epsilon^2\right)^2\mathbb{V}_{\mathcal{P}}[\mathcal{L}(\mathcal{Z})]}.$$

We have

$$
\mathbb{E}_{\mathcal{Q}}[\mathcal{L}(\mathcal{Z})] \leqslant \left(1 - \epsilon^2\right)^2 \mathbb{E}_{\mathcal{P}}[\mathcal{L}(\mathcal{Z}) - M] + M + 2\left(1 - \epsilon^2\right)\sqrt{\left(1 - \left(1 - \epsilon^2\right)^2\right)\mathbb{V}_{\mathcal{P}}[\mathcal{L}(\mathcal{Z})]}
$$

$$
- \frac{\left(1 - \left(1 - \epsilon^2\right)^2\right)\mathbb{V}_{\mathcal{P}}[\mathcal{L}(\mathcal{Z})]}{\mathbb{E}_{\mathcal{P}}[M - \mathcal{L}(\mathcal{Z})]}
$$

$$
= \mathbb{E}_{\mathcal{P}}[\mathcal{L}(\mathcal{Z}) - M] + (1 - \epsilon^2)^2 \mathbb{E}_{\mathcal{P}}[\mathcal{L}(\mathcal{Z}) - M] - \mathbb{E}_{\mathcal{P}}[\mathcal{L}(\mathcal{Z}) - M] + M + 2\lambda_\epsilon \sqrt{\mathbb{V}_{\mathcal{P}}[\mathcal{L}(\mathcal{Z})]}
$$

$$
- \frac{\epsilon^2 \left(2 - \epsilon^2\right)\mathbb{V}_{\mathcal{P}}[\mathcal{L}(\mathcal{Z})]}{\mathbb{E}_{\mathcal{P}}[M - \mathcal{L}(\mathcal{Z})]}
$$

$$
= \mathbb{E}_{\mathcal{P}}[\mathcal{L}(\mathcal{Z})] + 2\lambda_\epsilon \sqrt{\mathbb{V}_{\mathcal{P}}[\mathcal{L}(\mathcal{Z})]} + \epsilon^2\left(2 - \epsilon^2\right)\left[M - \mathbb{E}_{\mathcal{P}}[\mathcal{L}(\mathcal{Z})] - \frac{\mathbb{V}_{\mathcal{P}}[\mathcal{L}(\mathcal{Z})]}{\mathbb{E}_{\mathcal{P}}[M - \mathcal{L}(\mathcal{Z})]}\right],
$$

where $\lambda_\epsilon = \sqrt{\epsilon^2\left(2 - \epsilon^2\right)\left(1 - \epsilon^2\right)^2}$.

Finally, we get the upper bound of $\mathbb{E}_{\mathcal{Q}}[\mathcal{L}(\mathcal{Z})]$:

$$
\mathbb{E}_{\mathcal{Q}}[\mathcal{L}(\mathcal{Z})] \leqslant \mathbb{E}_{\mathcal{P}}[\mathcal{L}(\mathcal{Z})] + 2\lambda_\epsilon \sqrt{\mathbb{V}_{\mathcal{P}}[\mathcal{L}(\mathcal{Z})]} + \epsilon^2\left(2 - \epsilon^2\right)\left[M - \mathbb{E}_{\mathcal{P}}[\mathcal{L}(\mathcal{Z})] - \frac{\mathbb{V}_{\mathcal{P}}[\mathcal{L}(\mathcal{Z})]}{\mathbb{E}_{\mathcal{P}}[M - \mathcal{L}(\mathcal{Z})]}\right].
$$

The above expression, incorporating the model $h_\theta$ as well as $x$ and $y$, can also be represented as:

$$
\mathbb{E}_{\mathcal{Q}}[\mathcal{L}(h_\theta(x), y)] \leqslant \mathbb{E}_{\mathcal{P}}[\mathcal{L}(h_\theta(x), y)] + 2\lambda_\epsilon \sqrt{\mathbb{V}_{\mathcal{P}}[\mathcal{L}(h_\theta(x), y)]}
$$

$$
+ \epsilon^2\left(2 - \epsilon^2\right)\left[M - \mathbb{E}_{\mathcal{P}}[\mathcal{L}(h_\theta(x), y)] - \frac{\mathbb{V}_{\mathcal{P}}[\mathcal{L}(h_\theta(x), y)]}{\mathbb{E}_{\mathcal{P}}[M - \mathcal{L}(h_\theta(x), y)]}\right].
$$

**Proof of the conditions for the inequality to hold.** From right side of inequality (2) we have

$$
\frac{S}{(1 - S^2)^{\frac{1}{2}}} \geqslant \frac{\Delta F}{\langle f\psi_{\mathcal{P}}, \psi_{\mathcal{P}}\rangle} \implies \frac{|\langle\psi_{\mathcal{P}}, \psi_{\mathcal{Q}}\rangle|^2}{1 - |\langle\psi_{\mathcal{P}}, \psi_{\mathcal{Q}}\rangle|^2} \geqslant \frac{\mathbb{V}_{\mathcal{P}}[\mathcal{L}(\mathcal{Z})]}{\mathbb{E}_{\mathcal{P}}^2[M - \mathcal{L}(\mathcal{Z})]}.
$$

Thus we can obtain the condition of Lemma.1:

$$
\sqrt{\frac{|\langle\psi_{\mathcal{P}}, \psi_{\mathcal{Q}}\rangle|^2}{1 - |\langle\psi_{\mathcal{P}}, \psi_{\mathcal{Q}}\rangle|^2}} \geqslant \frac{\sqrt{\mathbb{V}_{\mathcal{P}}[\mathcal{L}(\mathcal{Z})]}}{\mathbb{E}_{\mathcal{P}}[M - \mathcal{L}(\mathcal{Z})]} \implies \sqrt{\frac{1 - |\langle\psi_{\mathcal{P}}, \psi_{\mathcal{Q}}\rangle|^2}{|\langle\psi_{\mathcal{P}}, \psi_{\mathcal{Q}}\rangle|^2}} \leqslant \frac{M - \mathbb{E}_{\mathcal{P}}[\mathcal{L}(\mathcal{Z})]}{\sqrt{\mathbb{V}_{\mathcal{P}}[\mathcal{L}(\mathcal{Z})]}} \implies
$$

$$
\frac{1}{\langle\psi_{\mathcal{P}}, \psi_{\mathcal{Q}}\rangle^2} \leqslant 1 + \left(\frac{M - \mathbb{E}_{\mathcal{P}}[\mathcal{L}(\mathcal{Z})]}{\sqrt{\mathbb{V}_{\mathcal{P}}[\mathcal{L}(\mathcal{Z})]}}\right)^2 \xrightarrow{\langle\psi_{\mathcal{P}}, \psi_{\mathcal{Q}}\rangle = 1 - \epsilon^2} \frac{1}{(1 - \epsilon^2)^2} \leqslant 1 + \left(\frac{M - \mathbb{E}_{\mathcal{P}}[\mathcal{L}(\mathcal{Z})]}{\sqrt{\mathbb{V}_{\mathcal{P}}[\mathcal{L}(\mathcal{Z})]}}\right)^2 \implies
$$

$$
1 - \epsilon^2 \geqslant \left[1 + \left(\frac{M - \mathbb{E}_{\mathcal{P}}[\mathcal{L}(\mathcal{Z})]}{\sqrt{\mathbb{V}_{\mathcal{P}}[\mathcal{L}(\mathcal{Z})]}}\right)^2\right]^{-\frac{1}{2}} \implies \epsilon^2 \leqslant 1 - \left[1 + \left(\frac{M - \mathbb{E}_{\mathcal{P}}[\mathcal{L}(\mathcal{Z})]}{\sqrt{\mathbb{V}_{\mathcal{P}}[\mathcal{L}(\mathcal{Z})]}}\right)^2\right]^{-\frac{1}{2}}.
$$

$$
\epsilon^2 \leq 1 - \left[1 + \left(\frac{M - \mathbb{E}_{\mathcal{P}}[\mathcal{L}(\mathcal{Z})]}{\sqrt{\mathbb{V}_{\mathcal{P}}[\mathcal{L}(\mathcal{Z})]}}\right)^2\right]^{-\frac{1}{2}}.
$$

That is:

$$
\epsilon^2 \leq 1 - \left[1 + \left(\frac{M - \mathbb{E}_{\mathcal{P}}[\mathcal{L}(h_\theta(x), y)]}{\sqrt{\mathbb{V}_{\mathcal{P}}[\mathcal{L}(h_\theta(x), y)]}}\right)^2\right]^{-\frac{1}{2}}.
$$

$\square$

The estimation of $\mathbb{E}_{\mathcal{P}}[\mathcal{L}(h_\theta(x), y)]$ and $\mathbb{V}_{\mathcal{P}}[\mathcal{L}(h_\theta(x), y)]$ is key in Lemma A.1. The estimation methods mentioned in Werber et al. (Weber et al., 2022) are shown below Proposition A.2 and A.3.

**Proposition A.2.** *(Hoeffding, 1963) (Hoeffding, 1963) Let $(x_1, y_1), \ldots, (x_n, y_n)$ be independent random variables drawn from $\mathcal{P}$ and taking values in $(x, y)$. Let $\mathcal{L} : (h_\theta(x), y) \to [0, M]$ be a loss function and let $\hat{\mathcal{L}}_n^2 := \frac{1}{n} \sum_{i=1}^n \mathcal{L}^2(h_\theta(x_i), y_i)$ be the mean under the empirical distribution $\hat{\mathcal{P}}_n$. Then for $\delta > 0$, with probability at least $1 - \delta$,*

$$\mathbb{E}_{\mathcal{P}}[\mathcal{L}(h_\theta(x), y)] \le \hat{\mathcal{L}}_n + M \left[ \frac{\ln(1/\delta)}{2n} \right]^{1/2}.$$

**Proposition A.3.** *(Maurer & Pontil, 2009) (Maurer & Pontil, 2009) Let $(x_1, y_1), \ldots, (x_n, y_n)$ be independent random variables drawn from distribution $\mathcal{P}$ and taking values in $(x, y)$. For a loss function $\mathcal{L} : (h_\theta(x_i), y_i) \to [0, M]$, let*

$$S_n^2 := \frac{1}{n(n-1)} \sum_{1 \le i < j \le n}^n \left( \mathcal{L}\left(h_\theta(x_i), y_i\right) - \mathcal{L}\left(h_\theta(x_j), y_j\right) \right)^2$$

*be the unbiased estimator of the variance of the random variable $\mathcal{L}(h_\theta(x), y)$, $(x, y) \sim \mathcal{P}$. Then for $\delta > 0$, with probability at least $1 - \delta$,*

$$\left[ \mathbb{V}_{\mathcal{P}}[\mathcal{L}(h_\theta(x), y)] \right]^{1/2} \le \left[ S_n^2 \right]^{1/2} + M \left[ \frac{2 \ln 1/\delta}{n - 1} \right]^{1/2}.$$

# B   Appendix B

**Case study.** To show how mismatches in parameter space and distribution space affect learning performance, we design a simple experiment (see Figure 5). We fine-tune a pre-trained Resnet18 model on a balanced subset of CIFAR10, resulting in a model $\theta_1$ that significantly improves classification performance. However, when fine-tuning it on an imbalanced subset of CIFAR10, the performance of model $\theta_1$ drops significantly, even though the parameter distance $\|\theta_1 - \theta_2\|_2^2$ remains the same.

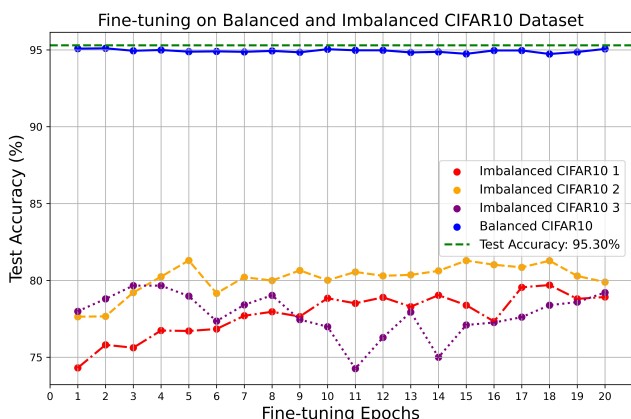

Figure 5: Although the distance between the models before and after fine-tuning remained consistent, their performance differed significantly.

Figure 6: we plot the distribution of the selected learners' teaching influence quantization scores.

Figure7: an illustration of class imbalance between CIFAR10 and CIFAR100.

Figure 8: we plot the score distribution using data indices as the variable.

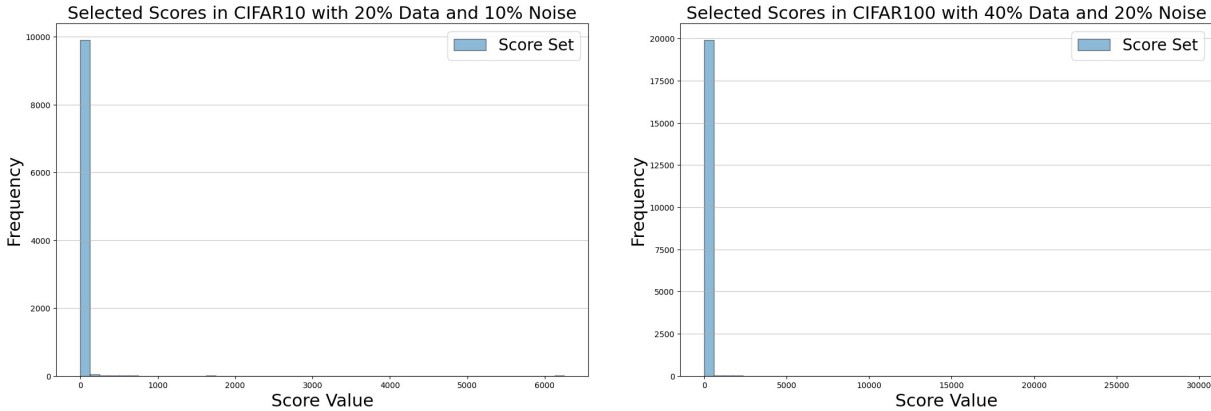

Figure 6: Score frequency distribution of the selected data, the left is a 20% subset under 10% noise for CIFAR10. The right is a 40% subset under 20% noise for CIFAR100.

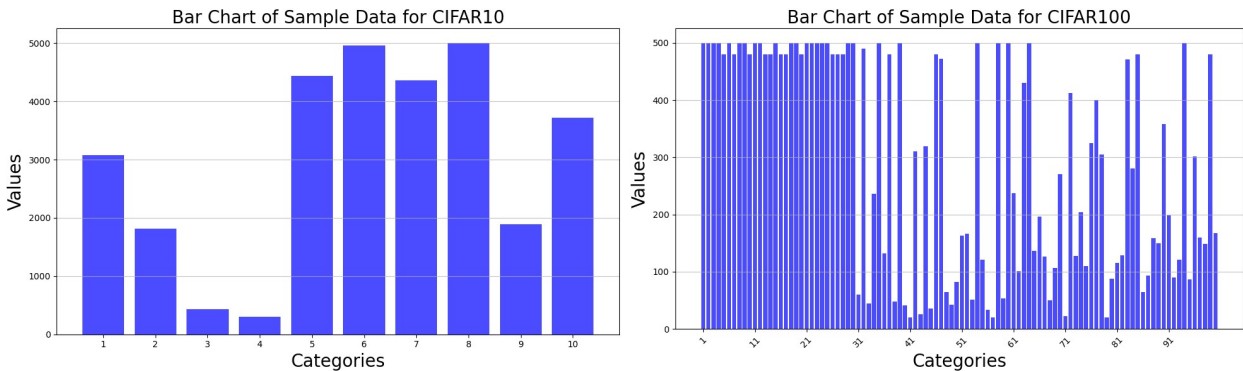

Figure 7: Class Imbalance for CIFAR10 and CIFAR100.

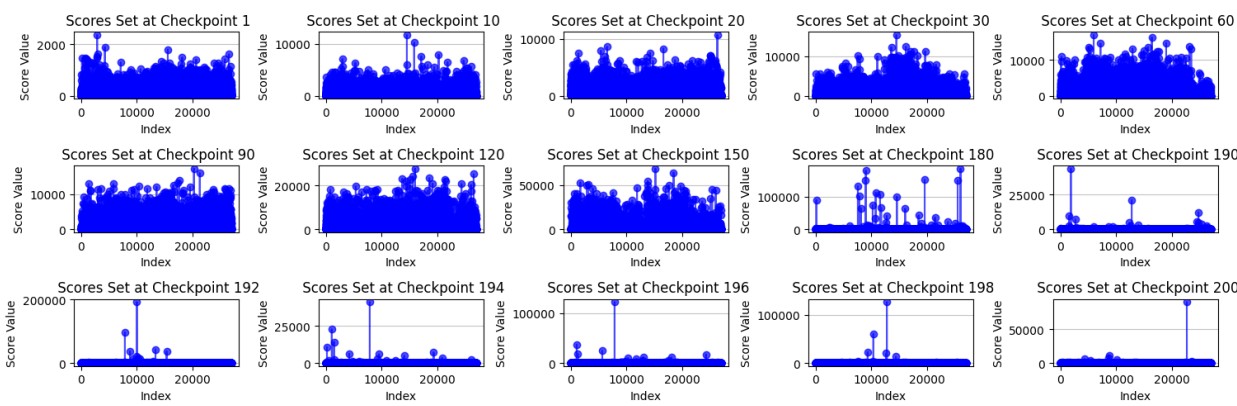

Figure 8: Score value plots, teaching influence score is obtained using 40% of the sample size on the imbalanced dataset of CIFAR10.

