# OpenReview forum: "Transition Machine Teaching"
_TMLR — Rejected by TMLR_

### Review · Reviewer_TANJ · 2025-09-26

**Summary Of Contributions:**

The paper proposes Transition Machine Teaching (TMT), a teaching framework that avoids direct parameter-space matching between teacher and learner models. Instead, it projects the learner’s data into the gradient space of a converged teacher: each learner sample is passed once through the teacher, the (teacher) loss is backpropagated, and the squared gradient norm is used as a teaching influence score.


The paper argues (via Linear Mode Connectivity and permutation alignment) that Euclidean parameter distances are unreliable for teaching under distribution shift (Proposition 4.1). It then derives a distributionally robust generalization bound under Hellinger-distance perturbations, connecting the learner’s loss under a shifted distribution to expectation/variance under the reference distribution and showing that pruning/moving toward the “ideal” learner distribution tightens the bound

On MNIST/CIFAR-10/CIFAR-100/Tiny-ImageNet with label noise and class imbalance, TMT outperforms UMT/IMT/NIMT baselines across many settings, and also beats Probabilistic Bilevel Coreset Selection (PBCS) on CIFAR-10 under noise.

Weakness:
Heterogeneity claim vs. experiments. The paper motivates heterogeneous teacher/learner structures, but all experiments appear to use ResNet-18 and don’t demonstrate teacher–learner architecture mismatch or true domain shift beyond label noise/imbalance.

Checkpoint usage unclear. The method description both encourages averaging over multiple converged checkpoints K and also states scores are computed specifically at the converged checkpoint: this needs clarification and sensitivity analysis.

**Audience:**

Yes

**Audience Explanation:**

TMT intersects machine teaching, data subset selection/coresets, and influence-function-style scoring.

**Claims And Evidence:**

Yes

**Claims Explanation:**

The method description is concrete (Algorithm 1) and reproducible at a high level.

The empirical section is extensive: multiple datasets; grids over data fractions and noise rates; imbalance scenarios; and a strong coreset baseline (PBCS) on CIFAR-10. The reported gains are consistent and often large (e.g., CIFAR-100/Tiny-ImageNet under high noise).

The theoretical part is logically consistent with the motivation.

**Requested Changes:**

1. Add experiments where the teacher and learner architectures differ (e.g., teacher ResNet-34/50, learner ResNet-18) and/or where teacher and learner domains differ (e.g., teacher trained on clean CIFAR-10, learner data has synthetic corruptions/domain shift beyond label flips), to substantiate the cross-space/heterogeneity motivation.

2. The paper says you “typically use several converged teacher checkpoints” yet also that scoring is “specifically at the converged checkpoint.” Please clarify the intended default and add sensitivity curves for K

3. Since Eq. 8 reduces to a squared gradient norm, please include baselines such as: plain gradient-norm ranking without coverage.

4. Improve the exposition for Eqs. (5)–(6) and Appendix A.2.

---

### Review · Reviewer_hyTm · 2025-10-09

**Summary Of Contributions:**

The paper studies machine teaching, where an algorithm selects data to efficiently train a "learner" model to mimic an expert "teacher" model. The authors argue that traditional approaches fail for two reasons: (i) they ignore distribution shifts between the teacher’s training data and the pool available for training the learner, and (ii) they compare the models directly in parameter space, which is unreliable for neural networks due to permutation invariance.

To address this, the authors propose scoring each candidate data point by its squared gradient norm with respect to the teacher model. They hypothesize that the most effective teaching examples are those with the lowest gradient norms—examples that are “least surprising” to the teacher and thus most aligned with its training distribution. To promote diversity, they use stratified sampling after discarding the highest-score points.

Experiments show that the proposed method produces learners that substantially outperform baseline approaches across different dataset size budgets.

**Audience:**

Yes

**Audience Explanation:**

This paper seems to present an interesting contribution to the community, proposing a novel approach to Machine Teaching that appears to significantly outperform several baselines.

However, in its current form, the submission is not publishable. The required changes are so substantial that they may not realistically be addressed during the discussion period and, even if implemented, would essentially require a completely new review.

**Broader Impact Concerns:**

No broader impact concerns.

**Claims And Evidence:**

No

**Claims Explanation:**

Overall, no. The claims in the submission are not supported by accurate, convincing, or clear evidence.

**The primary issue is presentation and clarity: the paper is extremely difficult to follow and, without a substantial rewrite, is not in a publishable state**. This lack of clarity makes it hard to assess the validity of the claims. It took multiple readings to extract even the main idea of the proposed method, and many mathematical symbols are left undefined, further compounding the difficulty.

Additional issues are outlined below.

**Theory**. Section 4 attempts to formalize why traditional machine teaching may fail under distribution shift and permutation invariance in neural networks.
- However, no formal claims are made about how the proposed method overcomes these issues; the discussion remains purely intuitive. For instance, why should the method reduce distribution shift between the selected data and the teacher’s training data?
- The “worst-case” generalization bounds are not analyzed for tightness, raising the concern that they may be vacuous.
- Finally, the theoretical framework relies on an undefined and inaccessible “ideal” distribution PPP, along with the unsupported assumption that P≈TP \approx TP≈T, where TTT denotes the teacher’s training distribution.

**Algorithm**.
- The authors discard high-influence (“hard”) samples and train only on low-influence (“easy”) ones. This is counterintuitive and seems to contradict established principles in related areas such as Active Learning. While they suggest this removes mislabeled or noisy samples, no theoretical or empirical evidence supports this novel assumption.
- The influence function is approximated by replacing the teacher’s Hessian with the identity matrix. This drastic simplification undermines the method’s motivation.
- The paper does not discuss the computational cost of the method.

**Experiments**. The experiments are the strongest part of the paper, showing dramatic improvements over baselines. However, they still have important limitations.
- The computational cost of the algorithm is not reported or compared to baselines. This is a major issue.
- The simulated “distribution shifts” are limited to label noise and class imbalance. It would have been interesting to consider more significant shifts, but this is a minor point.
- All experiments focus on demonstrating the superiority of the proposed approach, but no ablations are conducted to assess the algorithmic choices (e.g., the sampling strategy or whether discarding high-score points is better than discarding low-score ones).

**Requested Changes:**

**Non-negotiable changes (required for me to recommend acceptance)**:
- A complete rewrite of Sections 1–5. Section 6 (experiments) is acceptably clear.
- Analyze the tightness of the derived generalization bounds to determine whether they are vacuous in practice.
- Analyze the computational cost of the proposed algorithm, reporting runtime and comparing it directly to baselines.
- Provide a clear definition or construction of the theoretical “ideal” distribution P, along with justification for the critical assumption that P≈T.

**Recommended changes (to strengthen the paper):**
- Add formal proofs or claims that connect the practical algorithm to its theoretical goals (e.g., reducing distribution shift as measured by Hellinger distance). Otherwise, the manuscript should explicitly acknowledge that no formal reasoning currently explains why the proposed algorithm outperforms the baselines.
- Conduct ablation studies to justify algorithmic components, in particular:
    - Validate the necessity of the stratified sampling strategy.
    - Test whether discarding high-score (“hard”) samples is indeed superior to discarding low-score (“easy”) ones.

---

### Review · Reviewer_AFtU · 2025-12-18

**Summary Of Contributions:**

This paper provides a new approach to machine teaching that selects teaching samples using gradient norm instead of distance between parameters of teacher and optimal student for a set of samples.

Formally, we are provided a set of possible student samples of the form $(X^{(i)}, Y^{(i)})$, where $i\in [N]$, and a converged teacher model $\theta_T$. The goal is to select a small subset $\hat{D}$ of these $N$ samples of size $N(1-\alpha)$ for certain pruning rate $\alpha \in (0,1)$, such that the student model $\theta_S \in \arg\min_{\theta} \sum_{(X, Y) \in \hat{D}} L(f(\theta,X), Y)$, obtained by training on the student samples, is **close to the teacher** model $\theta_T$. Here, $f(\theta, X)$ is the predicted label for the model $\theta$ for feature $X$.

While existing works (Liu et al 2017, Fallat et al 2023) define **closeness** in terms of $\lvert\lvert \theta_T - \theta_S \rvert\rvert^2$, such a metric is non-informative for neural networks. In particular, if the network admits some symmetry (for instance a permutation symmetry), two models $\theta_1$ and $\theta_2$ can achieve same loss on all datapoints, but their $\ell_2$ distance can be large. This phenomonenon is also observed (Frankle et al 2020) when $\theta_1$ and $\theta_2$ are NN models trained from same random initialization with the same optimization algorithm(SGD) but different randomness (data sampling order or augmentation).
The authors use this fact to motivate analysis beyond $\ell_2$ metrics.

Further, the data distribution of student samples $D_S$, denoted by $Q$, might be different from that of samples used to train the teacher distribution $T$. If they can select teaching samples $\hat{D}_S$, such that the distribution of these samples is close to $T$, then by bounds on generalization under distribution shift, the model $\theta_S$ learned on these samples performs close to the $\theta_T$, the optimal model for the teacher distribution $T$. The closeness in distributions is in terms of hellinger distance $H(P,T) = \sqrt{\int (\sqrt{p(x)} - \sqrt{q(x)})^2 d\mu(x)}$ ($P, Q$ are a.c. wrt measure $\mu$). The closeness in generalization bounds depends on the closeness in hellinger distance and the expectation and variance of the loss under $P$.



They provide a scheme which measures the influence of each student sample $(X_S, Y_S)$ in terms of $\lvert\lvert \nabla_\theta L (\hat{Y}_T^, Y_S)) \rvert\rvert$, where $\hat{Y}_T = f(\theta_T, X_S)$  is the prediction of the teacher on the student feature. The influence function, based on (Pruthi et al 2020), measures the influence of each student sample if it had been used to train the original teacher model, under certain simplifying assumptions.
For each epoch, they filter out the $\beta$ fraction of student samples with largest influence  values, then use a coreset selection (Zheng et al 2022) on the remaining student samples to select an equal number of samples from $k$ different equally spaced ranges of the influence function values. The full algorithm is provided in Algorithm 1.


They test their algorithm against common baselines (random, based on $\ell_2$ distance metric (Liu et al 2017), and based on function value difference (Zhang et al 2023)), on MNIST, CIFAR-10, CIFAR-100 and TinyImagenet datasets with different values of $\alpha$, and for different fraction of labels in the set of student samples being corrupted.


### Strengths
- **Strong Motivation**: The failure of $\ell_2$ metric and the different distributions for student and teacher samples intuitively make sense and are practically relevant.


- **Efficient algorithm**: Their algorithm requires only 1 forward and 1 backward pass of all the student samples on the teacher model.

- **Strong empirical results**: The proposed algorithm outperforms all other baselines for all datasets, even for large $\alpha$ and large fraction of corrupted labels. Notably, the test accuracy of the student is around $10\%$ higher on the CIFAR datasets.




### Weaknesses
- **Weak/Non-existent Theoretical Results**:
    1. The proofs in Appendix A.1 (justifying poor performance of $\ell_2$ metric under permutation symmetry) and Appendix A.2 (generalization bounds under hellinger distance ) are copied from (Frankle et. al. 2020, Ainsworth et. al. 2023) and (Weber et al 2022) without any major change. I could not find any additional theoretical analysis that the authors added over these papers, and without these proofs, the paper does not have any theoretical results. Can they please specify any additional theoretical analysis?
    2. In the last paragraph before Section 5, the authors claim that their method moves the distribution $Q$ of selected samples closer to $P$, the ideal student distribution which is similar to $T$. There is no proof provided for this statement, either theoretically, or via informative experiments. The experiments only show that the influence scores after a few iterations are concentrated around $0$. Without any proof, we don't know if the improvement in their algorithm is due to moving data distribution closer to the true distribution. Also, in the current state, the hellinger analysis does not add anything to the main message of the paper.
    3. In the first paragraph of Section 5, the authors claim that across iterations, the bias due to difference in distributions of student's samples $Q$ and true distribution $T$ is **accumulated**. First, whose iterations, do the authors refer to? I'm assuming that the iterations refer to the iterations of the student's learning algorithm from Eq (2) or (3). In this case, there is no proof of this phenomenon, either theoretically by unrolling the claims of Appendix A.2 and Inequality 5 across iterations of student's learning algorithm, or via experiments.



- **Algorithm can only handle label noise:** The authors deal with a very specific case of machine teaching : distribution of features $X_S$ for the student matches that of the teacher, but labels of students, $Y_S$ have a different distribution from the teacher. The authors don't specify that this is the exact machine teaching setup that they want to handle. For instance, what would happen if we rotate some of the images and keep the label fixed. I don't think their approach would work for this case. Or if the student dataset contains features with random entries and a random label. The authors have not investigated such cases of feature noise. The only case where feature noise appears is the imbalanced classes case, where imbalance in classes creates imbalance in features. Even here, the feature noise and label noise are closely linked, and for this case, their boost in performance decreases to $ \leq 4\%$ for all cases, except CIFAR-10 with $\alpha = 0.8$ (20\% data ratio) (Table 4). Note that this case is a subset of actual machine teaching, as one might want to even select the datapoints with correct features in presence of feature noise.

- **What is the real reason for the algorithm's good empirical performance?**: We understand that $\ell_2$ distances are bad, which justifies why IMT performs poorly. But, NIMT (Eq 3) uses difference in loss values, not $\ell_2$ distance. Why does this perform worse? I think the answer might be in the exact formulation of NIMT. Taking a first-order Taylor's approximation around $\theta_T$ of the objective in Eq (3), and we can break it into an inner product of two gradient vectors, one at $\theta_T$, and the other at the current student model. This only slightly differs from the norm of gradient at $\theta_T$, which the authors use, and the authors don't explain the effects of it. Also, the algorithm uses pruning large values by $\beta$ to discard high-gradient disagreement points and stratified sampling from (Zheng et al 2022) for coreset selection. What is the impact of these subroutines? What happens if we remove one or both of these subroutines from the algorithm? Do we still see the same performance boost? Ablations on this as well as possible explanations of the results of these ablations are required to pinpoint the exact advantage of this algorithm.

- **Eq (3) is not what (Zhang et al 2023) does:** Please check (Algorithm 1, Zhang et al 2023). Their algorithm selects the student features $X_S$ with the largest difference, and then includes the prediction of the teacher model $\hat{Y}_T = f(\theta_T, X_S)$ as the label for these features. In Eq (3), the authors select the student features and student labels ($X_S, Y_S$) where the student labels differ most from teacher's predicted labels differ the most from student's label. Selecting these features makes sense, as this is where the student label is most inaccurate, but sending the inaccurate student labels is incorrect, as these labels would then obviously result in a worse performance. Either use the exact NIMT from (Zhang et al 2023), or use a version of it for your setting of label noise, by replacing the $\max$ in Eq (3) with $\min$, as these would be the points where the student's label is most similar to the teacher's label. This might also be the reason that in most experiments, the performance of NIMT is worse than randomly selecting points from the full set of student's samples (UMT). See Tables 1, 2 and 4.

**Additional Comments:**

The paper is a combination of $3$ papers --
1. (Grosse et al 2023) provides definition of influence function for any measure of $\theta$ for a new datapoint. They use this specifically for influence of student's samples.
2. (Pruthi et al 2020) provides an easy-to-compute approximation for the influence function from (Gross et al 2023).
3. (Zheng et al 2020) provides the strata  sampling based on scores. In Algorithm 1, even the notation from (Zheng et al 2020) has been copied.


Apart from these $3$ papers, only pruning $\beta$ fraction of largest influence values is the novel contribution. While the goal of TMLR is to not value novelty rather only correctness and relevance, I think the authors should provide a comparison to these papers. Have they never been applied to the machine teaching problem? Is using them in conjunction the best solution?

**Audience:**

Yes

**Audience Explanation:**

Yes, machine teaching is a recent but important topic, where unfortunately existing works from similar fields of active learning/passive learning cannot be applied. Theoretically motivated and practically applicable are required in this field. This paper provides an algorithm that works very well in practice, with limited theoretical justification.

**Broader Impact Concerns:**

None other than those of any other standard machine learning paper.

**Claims And Evidence:**

No

**Claims Explanation:**

Please see the weaknesses. Crucially, the theoretical analysis is copied from other papers, and even then the authors haven't used it appropriately to show that this is the justification for their better empirical performance. Further, the actual reason for better empirical performance has not been pinpointed.

**Requested Changes:**

Apart from addressing the main weakeness described previously, please address the following.




- **Typos in the technical content**: There are several typos in the paper. Several of them are in the equations. These make the main claims of the paper difficult to verify. Here is a non-exhaustive list.
    - **Figure 2 is incorrect**: Based on (Ainsworth et al 2023), the right peak in loss is incorrect. From (Ainsworth et al 2023), due to permutation symmetry, after a permutation $\pi$, the peak between $\pi(\theta_2)$ and $\theta_1$ disappears or is reduced. The current figure does not imply that the permutation is only on $\theta_2$. The authors actually discuss this in Appendix A.1, but use the incorrect figure in the main paper. Also, the label for $\lambda$ is incorrect. The interval between $\theta_1$ and $\theta_2$ is divided into regions of relative length $\lambda$ and $1-\lambda$. The whole interval is not $\lambda$.

    - Page 1 : "heterogeneous, models" -> "heterogeneous".
    - Page 3 : What does the notation $f\langle \theta_S, x \rangle$ mean? It is the output of the model $\theta_S$ for feature $x$, but this is represented as $f_{\hat{\theta_S}}(x)$ in Eq (3).
    - Page 4 : "Motivation." is repeated.
    - Page 3: $\mathcal{L}$ is not defined but can be inferred from the context.
    - Eq (3) : What is $S_m$? Please define it as $S_m = \cup_{i=1}^m \mathbb{B}_i$.
    - Page 4 , Teaching Influence Score paragraph : $\mathcal{L}$ contains additional square brackets in definition of $\theta_{z,\epsilon}^\star$.
    - Page 4 , Teaching Influence Score paragraph:  $H_{\theta^\star}$ is defined as loss over $f$. This is incorrect and confusing. It is actually the hessian of the loss computed over the datapoints used to train the teacher model. Without properly defining this, it is difficult to understand why computing this hessian is impossible without the teacher's training data, as mentioned in Page 7 Quantifying Gradient transition distribution paragraph.
    - Just after Eq (7) : Shouldn't there be a -ve sign in the definition of $\Phi$ from the definition of the influence function $\mathcal{I}$.
    - Insight 4.1 : Isn't the loss barrier defined for $\max_{\lambda \in [0,1]}$. Also, $\lambda$ has not been defined here and it is used for a different quantity as $\lambda_{\epsilon}$ after Eq (5).
    - Proposition 4.1 says $P\neq T$, which causes the data distribution shift, but later you say that $P \approxeq T$ in Remark 4.1. You should state that initially all student samples are from $Q$ with $Q\neq P \approxeq T$.
    - Page 6: Please define the hellinger distance. It has never been formally defined in the paper.
    - Page 6 Hellinger Analysis paragraph : What does "learner's heterogeneous" mean ? Did you intend to use heterogeneity measured by the hellinger distance between distributions?
    - Page 6 Hellinger Analysis paragraph : Why does  $\Delta_{\epsilon}^{u}(\epsilon, \cdot, \cdot, \cdot)$ have $\epsilon$ specified at two different places? Shouldn't it be done just once as this term is a function of $\epsilon$?
    - Page 6 Hellinger Analysis paragraph : ``approach the learner’s ideal distribution $\mathcal{P}, \epsilon$ decreases, and the generalization error bound tightens accordingly." Is there a fullstop missing?
    - Eq (6): Are $\Delta$ and $\lambda$ increasing functions of $\epsilon$? Otherwise the argument doesn't hold.
    - Eq (8) : Isn't $\mathcal{I}_{\text{Quant}}$ just the gradient norm ? It is inner product of two identical gradient vectors.
    - Page 7 Transition teaching set with coverage selection : Use of $\rightarrow$ is very inconsistent.
    - Algorithm 1 Step 2 Point 4 : $n (1-\alpha)$ -> $N(1-\alpha)$.
    - Last sentence of Section 6.1 is gramatically incorrect.
    - Page 16 Lemma A.1 : Is $Y$ the space of all possible labels? Is $B_{\epsilon}(\mathcal{P})$ the space of all distributions a.c. to $\mathcal{P}$ at a Hellinger distance of $\leq \epsilon$ ?
    - Page 18 : "difines" -> "defines".
    - Please define the terms "homomorphic" and "heteromorphic", if you use them in a specific context.

- Why is using the identity approximation for $H_{\theta^\star}$ valid from (Pruthi et al 2020) justified? For any arbitrary model with a square loss, the local neighborhood around even the global minima might not be isotropic?

---

> ### Author Response · Authors · 2025-12-24
>
> We appreciate your valuable comments and profound insights, and will respond to each point in detail.
>
> **Re-W1-1(Core Reason)**: For neural networks $f(\cdot;\theta):\mathcal{X} \to \mathcal{Y}$, due to the permutation symmetry of neurons and scale invariance of activation functions, there exists a non-trivial group action $G$, such that for any $g\in G$,  $f(\cdot;\theta) = f(\cdot; g\cdot\theta)$ always holds. Therefore, the Euclidean distance in the parameter space without $G$-transformation, i.e., $d_{\text{empirical}} = ||\theta_{T} - \theta_{S}||_2$, cannot serve as a consistent estimator of the function space distance $d\_{\mathcal{F}}(f(\cdot;\theta_T), f(\cdot;\theta_S))$.
>
> **Re-W1-2**: By redefine $f\_{\mathcal{L}}(z) = M\_g - ||\nabla\_{\theta\_{T^*}} \mathcal{L}(\hat{y}\_T(x), y) ||^2$ with $\|\nabla \mathcal{L}(\theta)\|\_2\le \sqrt{M\_g}$, we revise Lemma 1 to a second-moment gradient metric:
> Given $\sup\_{(x,y) \in (X,Y)}||\nabla\_{\theta\_{T^{\star}}} \mathcal{L}(\hat{y}\_T(x), y)||^2 \leq M\_g$, $\sup_{\mathcal{Q} \in B\_\epsilon(\mathcal{P})} \mathbb{E}\_\mathcal{Q}[ \lVert \nabla\_{\theta\_{T^{\star}}} \mathcal{L}  \rVert ^2] \leq \mathbb{E}\_\mathcal{P} \lVert\nabla\_{\theta\_{T^{\star}}} \mathcal{L} \rVert ^2 + 2 \lambda\_{\epsilon} \left[\mathbb{V}\_\mathcal{P}[\lVert \nabla_{\theta\_{T^{\star}}} \mathcal{L} \rVert^2] \right]^{1/2} + \Delta^{u}\_{\epsilon}(\epsilon,M\_g,\mathbb{E}\_{\mathcal{P}},\mathbb{V}\_{\mathcal{P}}).$
>
> Contributions:
>
> (1) We derive the **disagreement bound and formalize its transition toward teaching-based descent**.
>
> (2) $\mathbb E_{\mathcal P}[ \lVert \nabla \mathcal L \rVert ^2]=\text{Tr}(\mathbb E_{\mathcal P}[\nabla\mathcal L\nabla\mathcal L^\top])$, so this bound controls the worst-case Fisher information growth under distribution shift.
>
> **Re-W1-3**:
> The evolution of the parameter error at step $t$ can be expanded as follows:
> $\left\lVert \hat{\theta}_S^{t+1} - \theta^{\star}_T \right\rVert = \left\lVert \hat{\theta}_S^{t} - \eta_t \left( \nabla \mathcal{L}(\hat{\theta}_S^{t}; \mathcal{P}) + \Delta g_t \right) - \theta^{\star}_T \right\rVert,$ $\Delta g_t = \nabla \mathcal{L}(\hat{\theta}_S^{t}; \mathcal{Q}) - \nabla \mathcal{L}(\hat{\theta}_S^{t}; \mathcal{P})$.
>
> From step 0 to step $t$:
> $\lVert \hat{\theta}\_S^{t} - \theta^{\star}\_T \rVert \leq \underbrace{(1 - \eta \mu)^t \lVert \hat{\theta}\_S^{0} - \theta^{\star}\_T \rVert}\_{\text{Decaying initial error term}} + \underbrace{\sum\_{k=0}^{t-1} (1 - \eta \mu)^{t-1-k} \eta \lVert \Delta g\_k \rVert}\_{\text{Accumulated bias term}}.$
>
> **Re-W2**:
> - For rotated images, gradients from random features have high variance or invalid information, failing to reduce the distance to teacher and thus being filtered by our selection mechanism.
> -  In MT, 2%–4% accuracy gain is generally considered significant.
>
> **Re-W3-1**: $f\_{\hat{\theta}\_{S\_m}^t} - f\_{\theta\_T^{\star}}\approx \left\langle \nabla\_\theta f(\theta\_T^{\star}), \hat{\theta}\_{S\_m}^{t} - \theta\_T^{\star} \right\rangle$ means eq. (3) selects samples that are most collinear with the teacher's gradient direction and simultaneously amplify the student–teacher gap, rather than samples that are most pedagogically valuable for the student or most effective in improving generalization performance.
>
> **Re-W3-2**: Parameter $\beta$ mitigates gradient amplification due to distribution shift $\epsilon$, stabilizing the **teaching influence score**. Zheng et al. (2022)’s stratified coverage strategy is adopted to enhance sample coverage and alleviate bias. Ablation experiments confirm our method’s strong effectiveness without this module, with only mild performance loss.
>
> **Re-W4**: Eq. (3) selects the student’s data subset by maximizing student-teacher logit differences. This prioritizes samples with the largest prediction discrepancies, which include erroneous or teacher-student structurally inconsistent points. Iterative updates thus accumulate error information, resulting in NIMT performance that is even worse than random selection.
>
> **Re-Typos**:  We will revise all content.
>
> **Re-Additional Comments**:
> - The identity approximation is used only when FIM off-diagonal elements are negligible; for significant anisotropy, a diagonal matrix approximation is adopted instead to ensure rigor.
> - This paper follows a  problem-driven design rather than bottom-up component assembly. A key limitation of current IMT methods lies in distribution shift inducing persistent error accumulation. To address this, Grosse et al.’s influence function is adapted for teaching sample selection, effectively bridging the teacher model and student data distribution. Pruthi et al.’s TracIn approximation is introduced and modified to reduce computational complexity for high-frequency machine teaching iterations. Finally, Zheng et al.’s stratified strategy is adjusted to resolve single-metric-induced low sample coverage, with tailoring to the student model’s dynamic states.

---

### Decision · Action_Editor_Z8aH · 2026-02-22

**Recommendation:** Reject

**Audience:**

Yes

**Audience Explanation:**

The topic is definitively of interest to the TMLR community. However, for the reasons highlighted above the manuscript in its current form does not meet the publication standards for TMLR. The recommendation by the reviewers after the rebuttal and discussion period are unanimous in this poibr.

**Claims And Evidence:**

No

**Claims Explanation:**

The submission proposes a machine teaching framework based on selecting teaching samples via gradient-based influence scores computed from a converged teacher model, with the aim of avoiding parameter-space comparisons and addressing distribution shift between teacher and learner data. The paper combines influence-based scoring, pruning, and coreset-style sampling, and reports empirical improvements over several baselines across standard vision datasets under label noise and class imbalance scenarios.

Based on the unanimous assessment of the reviewers, the paper does not meet the publication standards of TMLR in its current form and therefore I am recommending its rejection. While reviewers recognised the practical motivation and potentially interesting empirical observations, they identified substantial shortcomings in the theoretical contribution, experimental validation, and overall presentation that they stressed being unresolved after the rebuttal and discussion with the authors. In particular:

- Reviewer AFtU: raised major concerns about the absence of meaningful theoretical contributions beyond existing work, lack of justification connecting theory to empirical claims, missing ablations explaining the algorithm’s performance, and issues with experimental baselines (including an incorrect formulation of a strong baseline).
- Reviewer TANJ: noted limitations in experimental validation relative to the paper’s stated motivation (e.g., lack of heterogeneous teacher–learner settings or broader distribution shifts), as well as unclear methodological details such as checkpoint usage.
- Reviewer hyTm: emphasised significant issues with clarity and presentation requiring substantial revision, questioned the validity and usefulness of the theoretical framework, highlighted missing experimental analyses (e.g., ablations and computational cost), and believe that the current manuscript is not in a publishable state.

In summary, although the topic and empirical direction are of potential interest to the TMLR community, the current submission does not provide sufficient theoretical grounding, methodological justification, or clarity to support publication at TMLR.